# mRNA therapy restores euglycemia and prevents liver tumors in murine model of glycogen storage disease

Jingsong Cao[1,4], Minjung Choi[1,4], Eleonora Guadagnin[1], Maud Soty[2], Marine Silva[2], Vincent Verzieux[2], Edward Weisser[1], Arianna Markel [1], Jenny Zhuo[1], Shi Liang[1], Ling Yin[1], Andrea Frassetto[1], Anne-Renee Graham[3], Kristine Burke[3], Tatiana Ketova[3], Cosmin Mihai[3], Zach Zalinger[3], Becca Levy[3], Gilles Besin[3], Meredith Wolfrom[3], Barbara Tran[3], Christopher Tunkey[3], Erik Owen[3], Joe Sarkis [3], Athanasios Dousis[3], Vladimir Presnyak[3], Christopher Pepin[3], Wei Zheng[3], Lei Ci[3], Marjie Hard[3], Edward Miracco[3], Lisa Rice[1], Vi Nguyen[1], Mike Zimmer[1], Uma Rajarajacholan[1], Patrick F. Finn[1], Gilles Mithieux[2], Fabienne Rajas[2], Paolo G. V. Martini[1✉] & Paloma H. Giangrande [1✉]

Glycogen Storage Disease 1a (GSD1a) is a rare, inherited metabolic disorder caused by deficiency of glucose 6-phosphatase (G6Pase-α). G6Pase-α is critical for maintaining inter-prandial euglycemia. GSD1a patients exhibit life-threatening hypoglycemia and long-term liver complications including hepatocellular adenomas (HCAs) and carcinomas (HCCs). There is no treatment for GSD1a and the current standard-of-care for managing hypogly-cemia (Glycosade®/modified cornstarch) fails to prevent HCA/HCC risk. Therapeutic mod-alities such as enzyme replacement therapy and gene therapy are not ideal options for patients due to challenges in drug-delivery, efficacy, and safety. To develop a new treatment for GSD1a capable of addressing both the life-threatening hypoglycemia and HCA/HCC risk, we encapsulated engineered mRNAs encoding human G6Pase-α in lipid nanoparticles. We demonstrate the efficacy and safety of our approach in a preclinical murine model that phenotypically resembles the human condition, thus presenting a potential therapy that could have a significant therapeutic impact on the treatment of GSD1a.

[1] Rare Diseases, Moderna, Inc, Cambridge, MA, USA. [2] INSERM UMR1213, Université Claude Bernard Lyon 1, Lyon, France. [3] Platform, Moderna, Inc, Cambridge, MA, USA. [4] These authors contributed equally: Jingsong Cao, Minjung Choi. ✉email: Paolo.Martini@modernatx.com; phgiangrande@gmail.com

Glycogen storage diseases (GSDs) are a class of rare genetic disorders characterized by failure to synthesize or break-down glycogen due to enzyme abnormalities in glycogen metabolism[1,2]. Among them, glycogen storage disease type1a (GSD1a) (OMIM: 232200) is caused by the deficiency of the glucose-6-phosphatase-alpha (G6Pase-α, encoded by the *G6PC* gene), a key enzyme that catalyzes the last step in glycogenolysis and gluconeogenesis[3,4]. G6Pase-α is expressed in gluconeogenic organs, primarily in the liver, but also in the kidneys and small intestine[5]. GSD1a is characterized by severe hypoglycemia, since G6Pase-α plays a pivotal role at the junction between glycogen-olysis and gluconeogenesis[6]. The conversion of glucose-6-phosphate (G6P) to free glucose catalyzed by G6Pase-α is a key step in releasing glucose from the liver into the bloodstream; consequently, the absence of G6Pase-α causes GSD1a patients to suffer from life-threatening hypoglycemia during fasting[7]. As G6P is a metabolite at the crossroads of multiple metabolic pathways, accumulation of G6P leads to other metabolic imbal-ances such as lactic acidemia, hypertriglyceridemia, hyperur-icemia, hypercholesterolemia, and steatosis[8]. Furthermore, glycogen buildup in liver and kidneys leads to hepatomegaly and nephromegaly, which are hallmarks of GSD1a[7].

The current standard-of-care for GSD1a relies on vigilant dietary management[9]: frequent feedings (every 4–6 h) of uncooked or modified cornstarch[10] and gastric drip feeding of glucose through the night (mainly used in young patients)[11]. However, any feeding/ cornstarch interruptions or delays can result in serious complica-tions leading to death and dietary management alone is only par-tially effective at preventing the accumulation of glycogen and other underlying metabolic abnormalities that lead to long-term hepatic and renal complications[12]. Long-term hepatic complications include hepatocellular adenomas (HCAs) which, are observed in 75% of adult patients of which 10% are at risk of malignant transformation into hepatocellular carcinoma (HCC)[13–15]. The only curative treatment option for these patients is liver/kidney transplantation[16], which remains high-risk with long-term com-plications associated with chronic immunosuppression.

To circumvent the high-risk associated with liver transplantation for GSD1a patients, not to mention the challenges associated with finding matched donors, several less-invasive alternatives are being pursued. Liver stem cell infusion restores metabolic para-meters without complications, but its therapeutic effects are transient (lasting only for a few months)[17,18]. Several somatic gene therapies that use an array of viral vectors have shown some promise in correcting hypoglycemia and prevention of HCA in GSD1a animal models[19–22] and one of these is currently being evaluated in humans[23]. However, the clinical application of these approaches is likely to be limited by the gradual loss of transgene expression over time, the potential risk of genotoxicity, and preexisting neutralizing antibodies[24]. In addition, due to its highly hydrophobic nature and localization in the endoplasmic reticu-lum membrane, G6Pase-α poses considerable challenges for protein purification and drug delivery, thereby impeding enzyme replacement therapy (ERT) as an option for GSD1a[25].

Restoration of protein function via delivery of mRNA to tissues offers considerable advantages over conventional methods. This platform can encode for any protein sequence of choice and utilizes the intracellular machinery for the production and proper cellular localization of the target protein for therapeutic or pre-ventative benefit[26–30] (Fig. 1a, example provided for G6Pase-α enzyme). Unlike viral vector-mediated gene delivery approaches, mRNA therapy corrects for protein function without modifying the genomic DNA[31,32]. Furthermore, the mRNA-dependent transient protein expression mitigates the risk of unintentional overdose due to constitutive and/or prolonged activation of protein function and the linear dose response observed with

mRNA therapy may allow titrating an ideal dose for each patient, something not easily feasible with viral vector-mediated gene therapy[29,30]. Despite numerous benefits, the advancement of mRNA-based therapeutics in the clinic has been hampered by lack of efficient and safe delivery methods that can transport long chains of negatively charged nucleotides across the cellular membrane. Recent developments in the encapsulation of mRNAs in lipid nanoparticles (LNPs) as delivery vehicles have enabled several proof-of-concept preclinical and clinical studies[33,34]. Furthermore, advances in mRNA chemistries have greatly improved safety profiles for non-immunostimulatory mRNA-based therapies[35–37]. The therapeutic potential of safe and effi-cacious re-dosing of our LNP technology has been demonstrated, in mouse models, for other liver metabolic diseases such as methylmalonic acidemia (MMA)[38,39], acute intermittent por-phyria (AIP)[40,41], Fabry disease[42], and others[43–45]. These pre-clinical findings are particularly encouraging as they suggest that the mRNA therapeutic modality can restore intracellular or transmembrane proteins, which are considered undruggable by current ERTs.

In this report we evaluate the efficacy and safety of mRNA therapy for GSD1a following repeat dosing. Previous efforts centered on developing an mRNA-based treatment for GSD1a were limited to single-dose, proof-of-concept studies performed in mouse models of the disease, to assess the effects of the exo-genously delivered mRNA on fasting blood glucose levels[46]. Importantly, given the need for chronic therapy for the treatment of this disease, herein we show that mRNA therapy can address both the life-threatening hypoglycemia, as well as the long-term high risk of HCA/HCC associated with this disease. Together, these results highlight the therapeutic potential of LNP encap-sulated mRNAs for GSD1a.

## Results

**Identification of optimized mRNA sequence encoding human G6Pase-α.** To ensure effective mRNA performance in vivo, we optimized protein sequences as well as codon choices in the mRNA sequence (Supplementary Table 1). We first performed a computer-aided bioinformatics search for consensus protein sequence and identified amino acid residues that are highly conserved among >100 mammalian orthologs (Fig. 1b). The top ten out of a total of 20 G6Pase-α protein variants derived from the bioinformatics analysis were individually evaluated for expression and enzymatic activity in HeLa cells. The G6Pase-α protein variant, bearing the serine (S) to cysteine (C) substitution at position 298 (S298C), showed an improvement in expression levels and activity by >2-folds compared to wild type human G6Pase-α (hG6Pase-α_WT) (Fig.1c). Therefore, we selected the S298C protein variant for further analysis. This finding is con-sistent with and supports previous studies by Zhang et al., which reported similar improvements in protein expression with the S298C variant[47,48]. Based on predicted topology analysis[49], the S298C substitution falls within the eighth transmembrane domain of hG6Pase-α, downstream of residues R83, H119, and H176[50], that are directly involved in hG6Pase-α activity (see predicted topology, Supplementary Fig. 1).

Next, we evaluated the subcellular localization of the exogenous hG6Pase-α_S298C variant protein. Co-localization of hG6Pase-α_S298C protein with calnexin (an ER marker) confirmed the proper ER subcellular localization of the variant (Fig. 1d, top panels) and matched the co-localization pattern of the G6Pase-α_WT protein indicating that the S to C substitution at position 298 does not interfere with localization to the ER membrane. Also, using Mander's colocalization coefficient analysis, we confirmed that the hG6Pase-α signals were significantly more

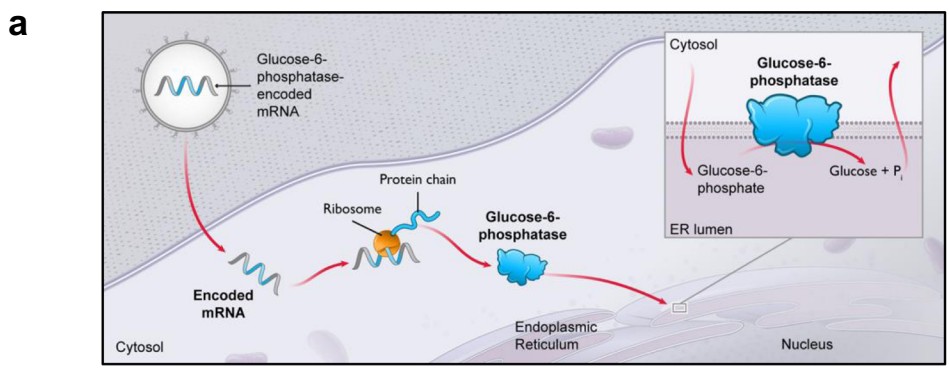

overlapped with the ER marker (Calnexin) signals than that of the mitochondrial maker (Tom20), which was used as a negative control (Fig. 1d, bottom panels). Together these data suggest that the modified h*G6PC* S298C mRNA expresses robust levels of hG6Pase-α S298C protein within the appropriate subcellular compartment (i.e., ER).

In previous work, we have demonstrated the benefit of codon optimization for maximizing protein expression and activity[38,40,51]. We have employed a similar approach to further enhance protein expression of hG6PC mRNA sequence. As shown in Fig. 2, codon-optimized (CO) h*G6PC* WT and S298C mRNAs resulted in overall higher expression and enzymatic

**Fig. 1 In vitro characterization of modified mRNA encoding hG6Pase-α. a** Hypothetical model of *hG6PC* mRNA therapy. *hG6PC* mRNAs are delivered to liver via lipid nanoparticles. Once the mRNA is in the cell (hepatocytes) it is translated by the cellular machinery into a functional protein that is localized to the ER membrane (likely following a co-translational translocation model), resulting in an active G6Pase-α enzyme. **b** Protein consensus screening by ortholog residue analysis. Top: WebLogo representation of the abundance of each alternative amino acid used at indicated residue positions. Bottom: The degree of conservation of amino acids at each position was quantified as relative entropy (Kullback–Leibler divergence). **c** Relative hG6Pase-α protein expression (solid circle) and hG6Pase-α enzymatic activity (solid square) in HeLa cells treated with the top ten *hG6PC* mRNA variants generated using protein consensus analysis. Data were shown as percentage of wild-type (WT) group and presented as mean ± SD of $n = 2$ (for protein expression), 3 (for enzymatic activity, Q247R), or 4 (for enzymatic activity, all other groups) biologically independent samples. **d** Subcellular localization of WT hG6Pase-α and S298C variant in HeLa cells. Green: hG6Pase-α, Red: Calnexin, an ER marker (top); TOM20, mitochondrial marker (bottom). Scale bars are 10 μm. The ratio of colocalized signal over total signal was calculated by Mander's colocalization coefficient analysis (bottom panel). Data were presented as mean ± SD of $n = 2$ biologically independent samples. Source data are provided as a Source Data File.

## a    *In vitro* (Hep3B cells)

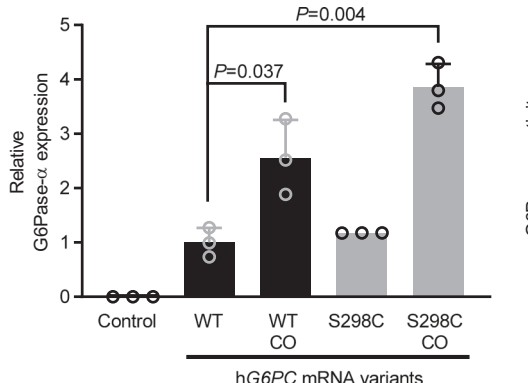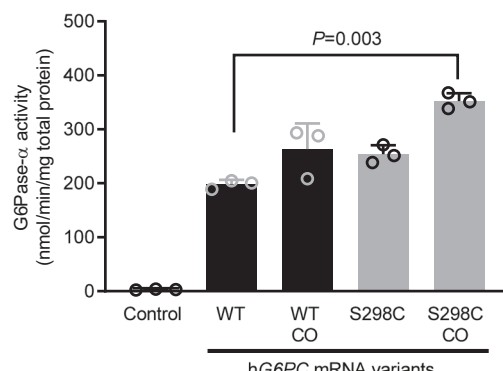

## b    *In vivo* (CD-1 mice)

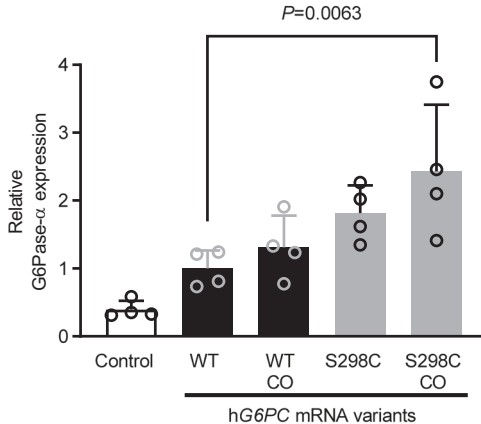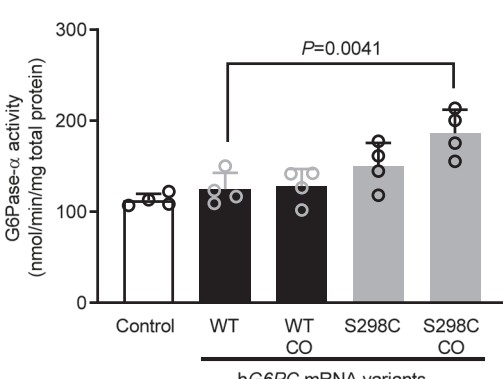

**Fig. 2 Effect of codon optimization on expression and activity of h*G6PC* mRNAs. a** hG6Pase-α protein expression (left panel) and enzymatic activity (right panel) of wild-type hG6PC (WT), codon optimized wild-type h*G6PC* (WT_CO), hG6PC_S298C (S298C), and codon optimized hG6PC_S298C (S298C_CO) mRNAs evaluated in Hep3B cells. Control cells were treated with eGFP mRNA. Data were presented as mean ± SD of $n = 3$ biologically independent samples. **b** hG6Pase-α protein expression (left panel) and enzymatic activity (right panel) of WT and codon optimized h*G6PC* mRNAs as evaluated in male CD-1 mice. Control animals were treated with eGFP mRNA. Data were presented as mean ± SD of $n = 4$ mice. For statistical analysis, raw values were Log2 transformed and subjected to one-way ANOVA, followed by the Dunnett's multiple comparisons test, compared to the non-codon optimized WT hG6PC mRNA. Statistically significant P values ($p \leq 0.05$) are shown in the graphs. Source data are provided as a Source Data File.

activity both in Hep3B cells (Fig. 2a) and in livers of WT (CD-1) mice (Fig. 2b). Of note, in CD-1 mice, the combination of protein variant S298C and codon optimization (S298C_CO mRNA) resulted in a more significant improvement in both hepatic protein expression and enzymatic activity (Fig. 2b).

**Hepatic h*G6PC* mRNA and hG6Pase protein half-lives.** To evaluate the impact of the CO h*G6PC* S298C mRNA (referred as

h*G6PC* S298C mRNA hereinafter) in livers, h*G6PC* S298C mRNA, protein and activity were measured over time. WT (CD-1) mice were i.v. administered with 1.0 mg/kg of eGFP, h*G6PC*-WT or -S298C mRNA-LNP and sacrificed at 6, 24, 72, 168, or 336 h ($n = 4$). Although the h*G6PC* S298C mRNA (transcripts, $T_{1/2}$: 20 h) was cleared rapidly from the liver (Fig. 3a), hG6Pase-α S298C protein expression ($T_{1/2}$: 79 h) peaked at 24 h and was detectable up to 168 h (7 days) (Fig. 3b). Similarly, enzyme activity ($T_{1/2}$: 74 h) was maximum at 24 h and continued for the lifetime of the

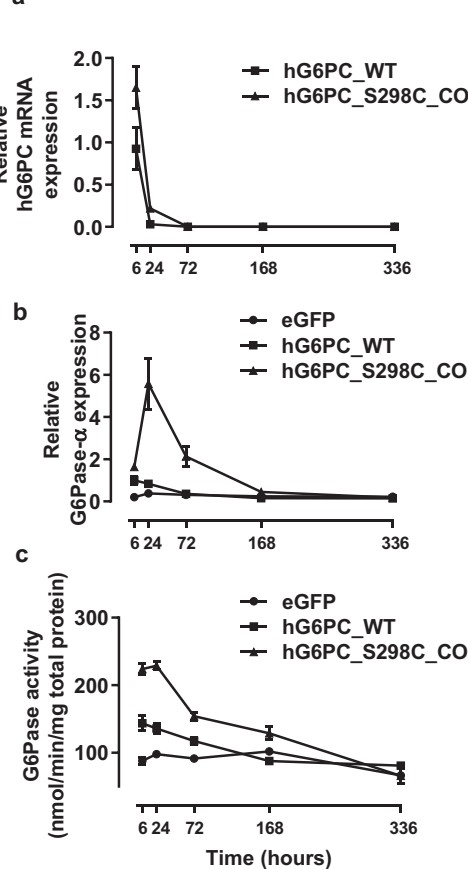

**Fig. 3 Hepatic hG6PC mRNA and hG6Pase-α protein and activity half-lives in wild-type mice.** Wild-type (CD-1) male mice were i.v. administered with 1.0 mg/kg of eGFP, h*G6PC*-wild type (WT), or codon-optimized *hG6PC*-S298C (hG6PC_S298C_CO) mRNA-LNP and sacrificed at 6, 24, 72, 168, 336 h (*n* = 4/group/sacrifice time point). **a** h*G6PC* mRNA levels (h*G6PC*-WT and S298C mRNAs). **b** Hepatic protein levels in mice treated with eGFP mRNA, mRNA encoding hG6Pase-α WT, or codon-optimized mRNA encoding hG6PC-S298C protein variant. **c** Hepatic enzymatic activity levels in mice treated with eGFP mRNA, mRNA encoding hG6Pase-α WT, or codon-optimized mRNA encoding hG6PC-S298C protein variant. Data were presented as mean ± SD (*n* = 3–4). Source data are provided as a Source Data File.

protein (Fig. 3c). Thus, the CO and protein engineered h*G6PC* mRNA sequence (S298C) resulted in higher hepatic hG6Pase-α protein levels and enzymatic activity compared to the WT mRNA. Despite the increase in hG6Pase-α hepatic protein levels and enzymatic activity, we did not observe any significant difference in the overall rate of clearance of the WT and S298C proteins (Fig. 3b, c).

**Efficacy of h*G6PC* mRNA-LNP in a liver-specific murine model of GSD1a (L.*G6pc*$^{-/-}$).** In vivo pharmacology assessments were performed in the liver-specific G6Pase-α null mouse model (L.*G6pc*$^{-/-}$) that recapitulates many of the disease hallmarks seen in GSD1a patients[52,53]. Like GSD1a patients, the L.*G6pc*$^{-/-}$ mice are unable to convert glycogen into glucose, leading to severe hypoglycemia upon fasting[52]. These mice also present with other hallmarks of GSD1a including, hepatomegaly, hepatic steatosis, hypertriglyceridemia, and as they grow older, HCAs and HCCs[53,54]. Additional details pertaining to the background and the genotype of the strain can be found in the methods section.

Initially, in a dose-ranging study, four groups of L.*G6pc*$^{-/-}$ mice (*n* = 5–10 per group) were injected i.v. with a single dose of either 1.0 mg/kg of eGFP mRNA, or 0.2, 0.5, or 1.0 mg/kg of h*G6PC* S298C mRNA. In addition, one control group of wild-type C57BL/6 J mice (WT) received phosphate-buffered saline (PBS). Fasting was initiated immediately after administration of the mRNAs, and blood glucose levels were monitored prior to mRNA administration/initiation of fast (0 h) and at 2.5-, 6-, and 24-h post-mRNA administration/initiation of fast. As shown in Fig. 4a, in contrast to eGFP mRNA treated mice, mice treated with h*G6PC* S298C mRNA showed significant improvement in fasting glycemia at all doses tested. Of note, blood glucose was above 60 mg/dL (therapeutic threshold based on clinical observations) in mice that received h*G6PC* S298C mRNA at all doses tested. While the increase in fasting blood glucose was dose-dependent at 2.5-h post-fasting, the fasting glucose levels observed at 6- or 24-h post-mRNA administration/initiation of fast did not increase with increasing doses, suggesting an adequate physiological regulation of blood glucose during fasting to maintain blood glucose around 100 mg/dL as in the WT mice (Fig. 4a). Mice were euthanized at 24-h post-mRNA administration to evaluate liver morphology (Fig. 4b, left panel), liver weight (Fig. 4b, right panel), hG6Pase-α protein (Fig. 4c, left panel and Supplementary Fig. 2, left panel) and enzymatic activity (Fig. 4c, right panel), and hepatic biomarkers including glucose-6 phosphate (G6P) (Fig. 4d, left panel), glycogen (Fig. 4d, middle panel and Supplementary Fig. 3, top panel), and triglycerides (Fig. 4d, right panel). As shown in representative liver images in (Fig. 4b), livers of eGFP mRNA-treated L.*G6pc*$^{-/-}$ mice were enlarged, pale, and steatotic in appearance as compared to livers from WT mice. In contrast, livers of h*G6PC* S298C mRNA treated L.*G6pc*$^{-/-}$ mice more closely resembled livers of WT mice. In addition, total liver weight was reduced for all three dose levels (Fig. 4b, right panel) which, as expected, correlated to a dose-dependent increase in hG6Pase-α protein levels (Fig. 4c, left panel and Supplementary Fig. 2, left panel) and enzymatic activity (Fig. 4c, right panel) as well as, an increase in h*G6PC* S298C mRNA in hepatocytes (Supplementary Fig. 3, bottom panel). Consistent with the above observations, treatment with h*G6PC* S298C mRNA at all three dose levels resulted in significant reduction in GSD1a hepatic biomarkers (glycogen, G6P, and triglycerides) compared to the eGFP-treated group (Fig. 4d and Supplementary Fig. 3, top panel). Treatment with h*G6PC* S298C mRNA also resulted in a robust decrease in serum triglycerides at all three dose levels tested, correlating with the reduction in hepatic triglycerides levels observed (Fig. 4e).

Next, a duration of action study was conducted in L.*G6pc*$^{-/-}$ mice to evaluate the effect of h*G6PC* S298C mRNA on fasting blood glucose. Blood glucose was monitored on days 0 (the day of administration), 2, 4, 7, 10, and 14 prior to (time 0) or at 2.5- or 6-h post-fasting, following a single administration of the h*G6PC* S298C mRNA. As shown in Fig. 5a, mice treated with the h*G6PC* S298C mRNA at doses ≥0.5 mg/kg showed statistically significant improvement in fasting blood glucose when compared to control mice treated with the eGFP mRNA on days 0, 2, and 4 post-administration. Such an improvement was also observed, at least partially, on days 7 and 10 post-administration (Fig. 5a). The 2.5- and 6-h fasting glucose levels in mice treated with h*G6PC* S298C mRNA were also maintained at above 60 mg/dL (therapeutic threshold), for at least 7 days (Fig. 5a). By day 14 post-mRNA administration, no significant difference in fasting blood glucose was observed between eGFP mRNA-treated and h*G6PC* S298C mRNA-treated groups (Fig. 5a). While the dosing regimen (weekly dosing) for maintaining euglycemia in the mouse model (Fig. 5a; as indicated by therapeutic threshold of 60 mg/dL or above) may seem impractical for clinical application, our

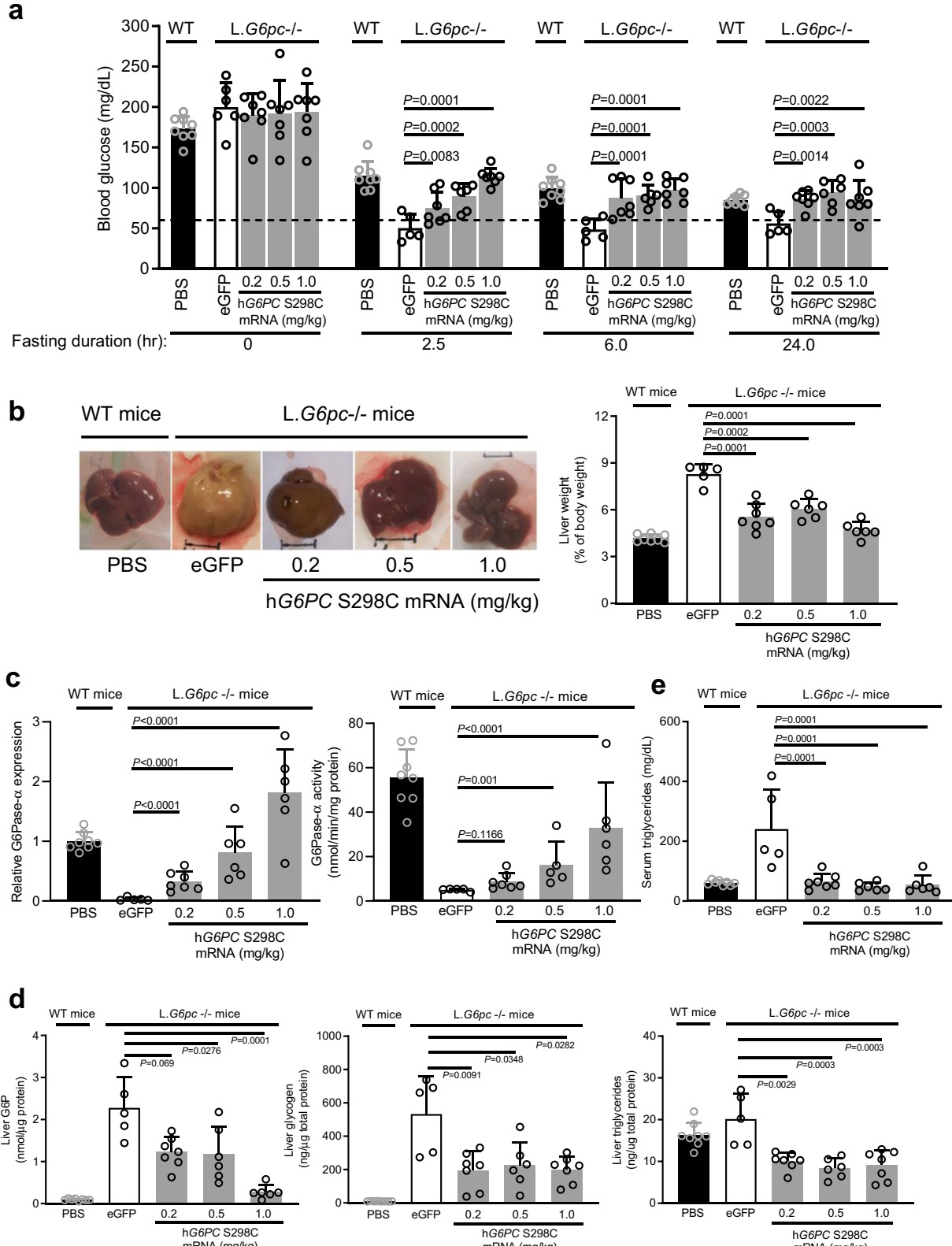

expectation is that the necessary dosing frequency will be substantially lower, likely at 3-week intervals or longer, due to slower drug metabolism in humans vs. mice as determined by allometric scaling[55–57].

Given the need of chronic administration of therapeutic h*G6PC* S298C mRNA to treat GSD1a patients, repeat dose studies were conducted to evaluate efficacy in the L.*G6pc*$^{−/−}$ mice

(Fig. 5b). L.*G6pc*$^{−/−}$ mice ($n = 7$–9 per group) received five consecutive injections of either eGFP mRNA or h*G6PC* S298C mRNA administered every 10 (second dose) to 14 days (all other doses) for over a period of 8 weeks. The mRNA was dosed at 0.25 mg/kg (Fig. 5b). Based on the single-dose efficacy studies (Figs. 4a and 5a), the 2.5-h post-fasting blood glucose level was determined to be equally predictive of efficacy as the longer 6-h post-fasting

**Fig. 4 Single i.v. dose of h*G6PC* S298C mRNA-LNP restores euglycemia, as well as serum and hepatic biomarkers in L.*G6pc*$^{-/-}$ mice. a** Blood glucose levels following administration of h*G6PC* S298C mRNA-LNP in L.G6pc$^{-/-}$ mice. WT, wild-type mice. (WT treated with PBS, $n = 8$ per group; L.G6pc$^{-/-}$ treated with eGFP, $n = 6, 5, 5$, and 5 per group for fasting duration of 0, 2.5, 6, and 24 h, respectively; L.G6pc$^{-/-}$ treated with h*G6PC* S298C at 0.2 mg/kg, $n = 7$ per group for all time points; L.G6pc$^{-/-}$ treated with hG6PC S298C at 0.5 mg/kg, $n = 7, 6$, and 6 per group for fasting duration of 0, 2.5, 6, and 24 h, respectively; L.G6pc$^{-/-}$ treated with hG6PC S298C at 1.0 mg/kg, $n = 7$ per group for all time points). Data were presented as mean ± SD. **b** Liver morphology (left panel) and liver weight (right panel) following administration of h*G6PC* S298C mRNA in L.*G6pc*$^{-/-}$ mice. Representative liver images are shown from $n = 8, 5, 7, 6$, and 6 mice per group from WT treated with PBS, L.*G6pc*$^{-/-}$ treated with eGFP, and L.*G6pc*$^{-/-}$ treated with hG6PC S298C mRNA at 0.2, 0.5, or 1.0 mg/kg, respectively. **c** hG6Pase-α S298C protein expression and enzymatic activity in livers of L.*G6pc*$^{-/-}$ mice. **d** Hepatic biomarker analysis following administration of h*G6PC* S298C mRNA-LNP in L.*G6pc*$^{-/-}$ mice. Liver G6P (left panel), liver glycogen (middle panel), liver triglycerides (right panel). **e** Serum triglycerides following administration of h*G6PC* S298C mRNA-LNP in L.*G6pc*$^{-/-}$ mice. h*G6PC* S298C mRNA-LNP dose range: 0.2, 0.5, and 1.0 mg/kg. For **b–e**, quantitative data were presented as mean ± SD ($n = 8, 5, 7, 6$, and 6 mice per group for WT treated with PBS, L. *G6pc*$^{-/-}$ treated with eGFP, and L.*G6pc*$^{-/-}$ treated hG6PC S298C mRNA at 0.2, 0.5, or 1.0 mg/kg, respectively). For statistical analysis, raw values were Log2 transformed and subjected to one-way ANOVA, followed by the Dunnett's multiple comparisons test, compared to the eGFP mRNA treated group. Statistically significant *P* values ($p \leq 0.05$) are shown in the graphs. Source data are provided as a Source Data File.

glucose level. Indeed, it has been suggested that a 5–6-h fast in mice may be comparable to an overnight fast in humans[58]. In the multidose study (Fig. 5b), we have chosen to monitor blood glucose at 2.5-h post-fasting on days 0, 1, 4, 7, and 10 following administration of the mRNA. Consistent with observations from single dose studies (Figs. 4a and 5a), h*G6PC* S298C mRNA-treated L.*G6pc*$^{-/-}$ mice had a pronounced improvement in fasting glycemia, in comparison with control mice receiving eGFP mRNA treatment (Fig. 5b). Importantly, the improvement in fasting glycemia with h*G6PC* S298C mRNA was sustained over the course of the treatment and did not diminish with repeat dosing (Fig. 5b).

**Evaluation of safety of h*G6PC* mRNA-LNP in a liver-specific murine model of GSD1a (L.*G6pc*$^{-/-}$).** Safety is a key consideration for the development of any chronic therapy. Initially, in order to demonstrate the abrogation of immune stimulation by mRNA through the use of modified nucleotides[35–37], we assessed serum cytokine levels for interferon gamma (IFNγ), interleukin-1beta (IL-1β), tumor necrosis factor alpha (TNFα), and interleukin-6 (IL-6) (Fig. 5c, left to right) in L.*G6pc*$^{-/-}$ mice that were euthanized 24 h post-mRNA treatment from our dose-ranging study (Fig. 5c). No increase in measured serum cytokines was observed at any dose level of h*G6PC* S298C mRNA tested. In addition to a lack of increase in cytokine levels, we also observed a tendency of improvement in liver enzymes (e.g., ALT) in the mRNA treated mice (Fig. 5d). Of note, in the repeat dose study (Fig. 5b) no significant increase in serum IFNγ, IL-1β, IL-6, and TNFα levels was observed in L.*G6pc*$^{-/-}$ mice treated with five consecutive doses of 0.5 mg/kg of h*G6PC* S298C mRNA (Fig. 5e). In the same study, we also measured antidrug antibodies (ADA). Importantly, no appreciable ADA response was observed in the serum of mice treated with h*G6PC* S298C mRNA (Fig. 5f). Finally, no observed hypersensitivity (changes in body temperature, altered breathing and ruffled fir), mortality, body weight, and changes in behavior (i.e., loss of appetite and distress) was observed in treated mice (data shown for body weight) (Fig. 5g). While additional safety studies performed in larger animal models (i.e., rats and nonhuman primates) are warranted for future clinical development, the above data suggest that h*G6PC* S298C mRNA may be well-tolerated under the conditions of these studies.

**Therapeutic impact of h*G6PC* mRNA-LNP in long-term GSD1a pathology.** Due to deregulated glucose homeostasis, over 75% of GSD1a patients develop long-term liver complications, such as HCAs[13–15]. HCA presents in patients over 25 years of age and in 10% of cases, HCA undergoes malignant

transformation to HCC[15]. Unfortunately, while strict compliance to dietary therapy can address the life-threatening symptoms of GSD1a, it is often only marginally effective at preventing HCA/HCC. To evaluate the effect of h*G6PC* S298C mRNA on prevention of HCA/HCC, we induced HCC in L.*G6pc*$^{-/-}$ mice by feeding them a high fat/high sucrose (HF/HS) diet using the protocol we have previously published[54]. We then treated the L. *G6pc*$^{-/-}$ mice, with ten doses (dosed at 0.25–0.5 mg/kg) of h*G6PC* S298C mRNA or control eGFP mRNA administered every 1–2 weeks. While only one WT mouse (out of a total of 21 WT mice) developed a lesion, ~58% of the control L.*G6pc*$^{-/-}$ mice (16 out of 26) fed a HF/HS diet developed visible (macroscopic) HCA/HCC lesions (Fig. 6a, left panel). Of note, several mice within this cohort developed more than one lesion per liver (Fig. 6a, middle panel). In contrast, treatment with h*G6PC* S298C mRNA resulted in significantly fewer mice with visible lesions (8 out of 34 or ~23%) (Fig. 6a, left panel) and significantly fewer visible hepatic lesions per mouse (Fig. 6a, middle panel). Finally, overall tumor burden (determined by summing the area of each HCA/HCC lesion per liver sample) was reduced in the h*G6PC* S298C mRNA treated group vs. the eGFP mRNA treated group (Fig. 6a, right panel). These observations were further confirmed by morphological (Fig. 6b, top panels) and histological analysis of the representative liver sections from each cohort (Fig. 6b, bottom panels). Moreover, HCA/HCC-related biomarkers (i.e., PKM2, β-catenin, and p62)[54] (Fig. 6c); and genes associated with cellular proliferation and HCA/HCC development (i.e., *Tgfb1, Glul,* and *Ctnnb1*)[54] were also partially reversed with h*G6PC* S298C mRNA treatment (Supplementary Fig. 4a). In addition, alpha fetoprotein (AFP)—a serum biomarker associated with GSD1a-related HCA/HCC development[54]—was also partially reduced upon treatment with h*G6PC* S298C mRNA (Supplementary Fig. 4b). Finally, as observed in the chronic dose study (Fig. 5b), treatment with h*G6PC* S298C mRNA showed a significant positive effect on fasting glycemia throughout the course of the treatment (Supplementary Fig. 4c, bottom panel). Collectively, these data suggest that chronic treatment of h*G6PC* S298C mRNA reduces the risk of HCA/HCC, a long-term complication with GSD1a by functional restoration of hepatic G6Pase.

## Discussion

This is the first evidence that repeat administration in model mice of an mRNA-based therapy for GSD1a that appears to be well-tolerated and efficacious at improving both fasting-tolerance and hepatic lesions. In this study, we engineered chemically-modified, CO mRNAs encoding hG6Pase-α and encapsulated them in LNPs to enable delivery to the liver. We show that the engineered mRNAs resulted in a hG6Pase-α enzyme with increased expression and enzymatic activity compared to non-optimized mRNA

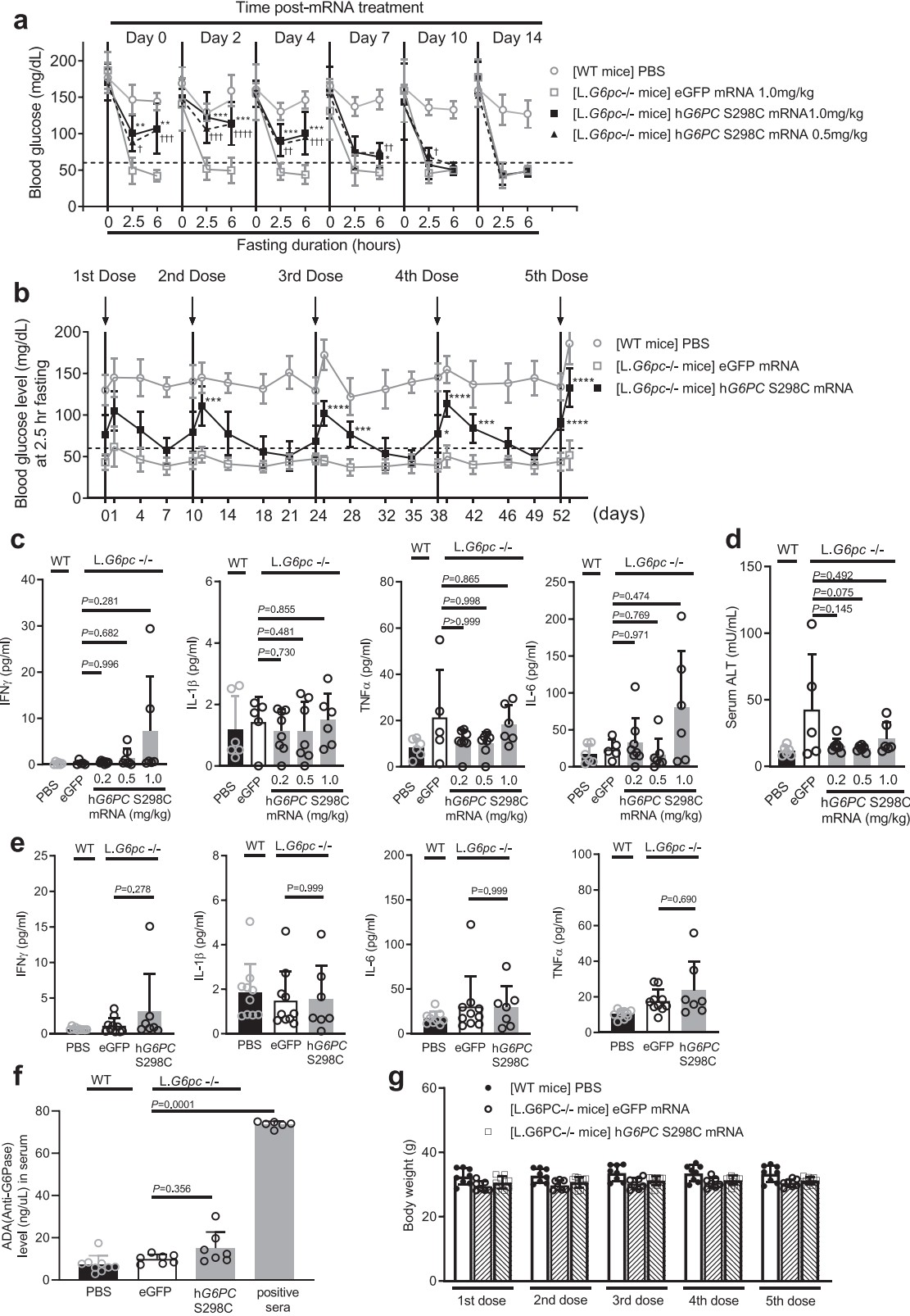

sequences (Fig. 1c). The protein translated from the engineered mRNAs was also directed to the proper subcellular compartment (endoplasmic reticulum) (Fig. 1d). When injected in a liver-specific mouse model of GSD1a that recapitulates the hepatic phenotypes in patients, the optimized mRNAs resulted in restoration of fasting blood glucose levels (Fig. 4a), normalization of several hepatic and serum biomarkers associated with GSD1a (Fig. 4b, d, e). Importantly, repeat administration of the mRNAs

**Fig. 5 Repeat i.v. dose of h*G6PC* mRNA-LNP results in safe and effective restoration of euglycemia in L.*G6pc*$^{-/-}$ mice. a** Single-dose duration of action of h*G6PC* S298C mRNA-LNP (0.5 or 1.0 mg/kg) administered i.v. in L.*G6pc*$^{-/-}$ mice. Blood glucose levels were measured at fed (0 h) or fasting conditions (2.5- or 6-h post-fasting). Data were presented as mean ± SD ($n = 8$, 9, 10, 10, and 10 mice per group for wild-type (WT) treated with PBS, L.*G6pc*$^{-/-}$ treated with eGFP, hG6PC S298C mRNA at 0.5 or 1.0 mg/kg, respectively). For statistical analysis, two-sample *t*-test (two-sided) was performed and corrected for multiple testing by using a Bonferroni adjusted level of 0.005. *$P \leq 0.05$, **$P \leq 0.01$, ***$P \leq 0.001$, ****$P \leq 0.0001$, comparing h*G6PC* S298C mRNA 1.0 mg/kg with eGFP ($p$ values are 0.0017 [day 0, 2.5 h], 0.0016 [day 0, 6 h], 0.0006 [day 2, 2.5 h], 0.0005 [day 2, 6 h], 0.001 [day 4, 2.5 h], 0.001 [day 4, 6 h], and 0.048 [day 7, 6 h], respectively). †$P \leq 0.05$, ††$P \leq 0.01$, †††$P \leq 0.001$, ††††$P \leq 0.0001$, comparing h*G6PC* S298C mRNA 0.5 mg/kg with eGFP ($p$ values are 0.033 [day 0, 2.5 h], 0.001 [day 0, 6 h], 0.0004 [day 2, 2.5 h], 0.00003 [day 2, 6 h], 0.002 [day 4, 2.5 h], 0.001 [day 4, 6 h], and 0.009 [day 7, 6 h], respectively) **b** Blood glucose levels following repeat (five doses) i.v. administrations of h*G6PC* S298C mRNA-LNP (0.25 mg/kg) in L.*G6pc*$^{-/-}$ mice. Arrows indicate dose administration. Blood glucose levels were measured at 2.5-h post-fasting. Data were presented as mean ± SD ($n = 8$, 7, and 9 mice per group for WT treated with PBS, L.*G6pc*$^{-/-}$ treated with eGFP, and L.*G6pc*$^{-/-}$ treated with h*G6PC* S298C mRNA, respectively). For statistical analysis, two-sample *t*-test (two-sided) was performed and corrected for multiple testing by using a Bonferroni adjusted level of 0.005. *$P \leq 0.05$, **$P \leq 0.01$, ***$P \leq 0.001$, ***$P \leq 0.0001$ comparing hG6PC S298C mRNA with eGFP ($p$ values are 0.005 [day 11], $4 \times 10^{-6}$ [day 25], 0.0007 [day 28], 0.0192 [day38], $1.3 \times 10^{-5}$ [day 39], 0.001 [day 42], $2 \times 10^{-6}$ [day 52], and $5 \times 10^{-5}$ [day 53], respectively). **c** Serum proinflammatory cytokines (from left to right): IFNγ, IL-1β, TNFα, and IL6 from the dose-ranging study. **d** serum ALT (mU/mL) levels from the dose-ranging study. For **c** and **d**, data were presented as mean ± SD ($n = 6$, 5, 8, 7, and 6 mice per group for WT treated with PBS, L.*G6pc*$^{-/-}$ treated with eGFP, and L.*G6pc*$^{-/-}$ treated with hG6PC S298C mRNA at 0.2, 0.5, or 1.0 mg/kg, respectively). **e** Serum proinflammatory cytokines (from left to right): IFNγ, IL-1β, TNFα, and IL6 from repeat-dose study. Data were presented as mean ± SD ($n = 10$, 10, and 7 mice per group for WT treated with PBS, L.*G6pc*$^{-/-}$ treated with eGFP, or hG6PC S298C mRNA). **f** Antidrug antibody assay measuring anti-G6Pase-α antibodies in sera of mice treated with five doses of h*G6PC* S298C mRNA-LNP (0.5 mg/kg). Data were presented as mean ± SD ($n = 9$, 7, 7, 6 mice per group for WT treated with PBS, L.*G6pc*$^{-/-}$ treated with eGFP, L.*G6pc*$^{-/-}$ treated with hG6PC S298C mRNA, and positive sera, respectively). **g** Body weight of L.*G6pc*$^{-/-}$ mice prior to each repeat i.v. dose treatment of h*G6PC* mRNA -LNP (0.25 mg/kg) for repeat dose study. Data were presented as mean ± SD ($n = 8$, 7, and 9 mice per group for WT treated with PBS, L.G6pc$^{-/-}$ treated with eGFP, and L.*G6pc*$^{-/-}$ treated with hG6PC S298C mRNA, respectively). For statistical analysis of **c–f**, raw values were Log2 transformed and subjected to one-way ANOVA, followed by the Dunnett's multiple comparisons test, compared to the eGFP mRNA treated group. *P* values are shown in the graphs (**c–f**). Source data are provided as a Source Data File.

was well-tolerated (Fig. 5e–g), resulted in the management of life-threatening hypoglycemia (Fig. 5b), may reduce the risk of long-term hepatic complications (e.g., HCA/HCC) (Fig. 6).

A key consideration when developing a drug for GSD1a is that the drug must have a sustained therapeutic effect. This can be achieved by (1) permanent gene correction (e.g., via gene editing), (2) gene insertion (e.g., vectored gene therapy), or (3) repeat administration of a transient therapeutic over the course of the patient's life. While gene editing approaches remain the holy grail for treating monogenic diseases, they are still in the early stages of development and several recent studies have raised concern about unintended consequences[59,60]. Notably, early versions of gene editing technology seem prone to hundreds (or even thousands) of unintended, off-target mutations throughout the genome[61]. Many of these mutations are likely to be silent and pose minimal risk to the patient, however, the risk of deleterious edits cannot be discarded offhand, especially germ line changes.

Gene therapy approaches have gained considerable momentum over the last 5 years[62,63], with several approvals granted by the US Food and Drug Administration (FDA) for treating various diseases, such as retinal dystrophy (LUXTURNA, Spark Therapeutics, Inc.), spinal muscular atrophy (ZOLGENSMA, AveXis, Inc.), and others[64]. Gene therapy has been successfully used to correct the pathologies associated with GSD1a in both mouse[19–22] and dog models[65], and is currently being evaluated in GSD1a adult patients[23]. However, clinical feasibility remains elusive primarily due to efficacy hurdles and the inherent risks associated with viral-based gene therapy. One limitation of gene therapy is known as the "dilution effect", where the therapy becomes less effective over time due to the natural process of liver cell growth and regeneration[66]. The dilution effect is the reason why young patients (<5 years old) are, generally, not eligible for gene therapy applications. Moreover, in GSD1a, hepatocyte proliferation is elevated, further compounding the issue[66]. In addition, the efficiency of AAV transduction is low (~5–10% of hepatocytes can be transduced) limiting transgene expression and correction. Thus, more than 90% of hepatocytes are not corrected and are prone to tumorigenesis. Finally, due to preexisting

neutralizing antibodies against viral vectors, which are present in >50% of the population (and likely higher in previously treated patients), repeat administration of AVV-based gene therapy is not ideal[67]. Thus, once treated, the patient may require a new vector serotype, eventually exhausting available options.

Systemic mRNA therapy offers several advantages as a therapeutic alternative to gene therapy, including (1) efficient protein expression without the need to first enter the nucleus, (2) essentially no integration risk, and (3) amenability to repeat dose. This third advantage is key, since, unlike the reported sustained, long-term expression associated with gene therapy, protein expression via mRNA-mediated delivery is transient and, like protein-based therapeutics (e.g., ERT), requires long-term chronic dosing. Here we show that our GSD1a mRNA therapy was well-tolerated and effective when dosed repeatedly, with little-to-no evidence of an immune response against the human protein in the GSD1a mouse model (Fig. 5e–g).

As discussed above, HCA/HCC is a long-term complication of GSD1a. The ability to restore G6Pase-α activity is expected to significantly reduce the risk of developing HCA/HCC overtime. Towards this end, previous studies performed in mouse models of GSD1a have shown that a recombinant adeno-associated virus (rAAV) vector-mediated *G6PC* gene transfer to either 2-week-old global *G6pc*$^{-/-}$ mice or adult L.*G6pc*$^{-/-}$ mice prevented HCA development[68,69]. While encouraging, the gene therapy approach was not able to abrogate preexisting tumors due to lack of expression of the virus in the adenoma lesions[69]. The authors went on to show that the viral transgene was under the control of glucocorticoid signaling which is impaired in the adenoma lesions, resulting in suppressed gene therapy mediated G6Pase-α restoration. Because mRNA therapy is not regulated at the level of transcription, this mechanism is not at play here. While purely speculative, the expectation is that GSD1a mRNA therapy may, not only prevent de novo HCA/HCC development at the tumor developing stage, but also potentially reduce any preexisting tumor burden. Given the potential of GSD1a mRNA therapy to impact preexisting tumors, further studies in older mice with preexisting adenomas are warranted.

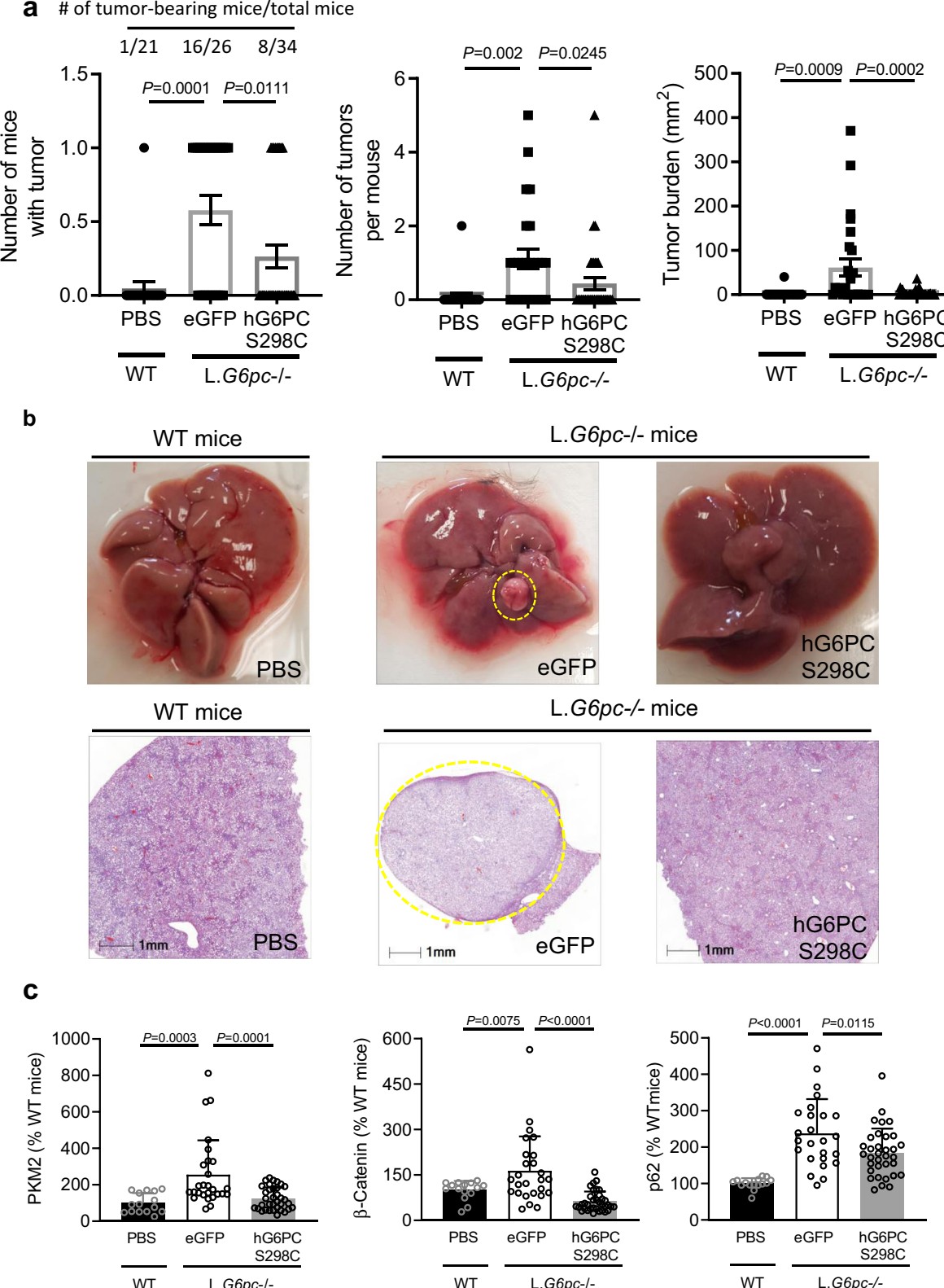

In summary, we describe a potential treatment for GSD1a which may address both the life-threatening hypoglycemia and HCA/HCC risk, that appears both well-tolerated and effective with repeat dosing under the conditions of these studies. mRNA therapy may overcome many of the limitations of the current standard-of-care for GSD1a (uncooked cornstarch and modified cornstarch—Glycosade®), as well as other treatment modalities. These data generated to date provide justification to continue developing mRNA therapy for the treatment of GSD1a. Finally, mRNA therapy has the potential to extend benefit to patients

**Fig. 6 Effect of h*G6PC* S298C mRNA-LNP on prevention of hepatic adenomas in L.*G6pc*$^{-/-}$ mice. a** Number of mice with tumors (left), number of tumors per mouse (middle), and tumor burden/area (right). Data were presented as mean ± s.e.m (*n* = 21, 26, and 34 mice per group for wild-type (WT) treated with PBS, L.*G6pc*$^{-/-}$ treated with eGFP, and hG6PC S298C mRNA, respectively). **b** Liver morphology (tumor-circled in yellow) (top panels) and liver histology (bottom panels) of WT and L.*G6pc*$^{-/-}$ mice treated with either eGFP mRNA or h*G6PC* S298C mRNAs. **c** HCA/HCC biomarkers (protein expression). Results are expressed as mean ± SD (*n* = 15, 27, and 35 mice per group for WT treated with PBS, L.*G6pc*$^{-/-}$ treated with eGFP, and L.*G6pc*$^{-/-}$ treated with hG6PC S298C mRNA, respectively). For statistical analysis, raw values were subjected to one-way ANOVA, followed by the Dunnett's multiple comparisons test, compared to the eGFP mRNA treated group. Statistically significant *P* values (*p* ≤ 0.05) are shown in the graphs. Source data are provided as a Source Data File.

suffering from other metabolic enzyme deficiencies of the liver that are not amenable to current treatment modalities.

## Methods

**mRNA production and formulation**. Complete N1-methylpseudouridine substituted mRNA was synthesized in vitro from a linearized DNA template containing the 5′ and 3′ untranslated regions (UTRs) and a poly-A tail, as previously described[37]. After purification, the mRNA was diluted in citrate buffer to the desired concentration and frozen. Complete sequence of h*G6PC*-S298C can be found in the Supplementary Materials (see Supplementary Table 1).

LNP formulations were prepared by ethanol drop nanoprecipitation as previously described[70]. Briefly, heptadecan-9-yl 8-((2-hydroxyethyl)(8-(nonyloxy)-8-oxooctyl)amino)octanoate, dipalmitoylphosphatidylcholine, cholesterol, and 1,2-dimyristoyl-glycero-3-methoxypolyethylene glycol-2000 were dissolved in ethanol and combined with acidified mRNA (sodium acetate, pH 5) at a ratio of 3:1 (aqueous:ethanol). Formulations were dialyzed against PBS, pH 7.4, in dialysis cassettes for at least 18 h. Formulations were concentrated using Amicon ultra centrifugal filters (EMD Millipore), passed through a 0.22 μm filter, and stored at 4 °C until use. All formulations were tested for particle size, RNA encapsulation, and endotoxin and were found to be suitable for in vivo use.

**Mammalian cell culture and transfection**. HeLa and Hep3B cells were obtained from ATCC and were maintained in DMEM media (10% FBS) at 37 °C supplied with 5% CO2. One day prior to transfection, 500,000 cells/well were seeded on 6-well plates, resulting in ~70% confluency on the day of transfection. Cells were transfected with 1 μg of mRNA using Lipofectamine 2000 or Messenger MAX™ (Invitrogen) by following the manufacturer's protocol. 24–48 h post-transfection, cells were harvested and used for protein expression or enzymatic activity measurement.

**Consensus sequence analysis for protein engineering**. Alternative G6Pase-α protein sequences in other mammalian species (nonhuman orthologs) were ranked by a score that maximizes the difference in relative entropy between the consensus substitution and the wild-type amino acid at that position. Relative entropy, also known as Kullback–Leibler divergence, is a common measure of amino acid conservation, defined at sequence position i for amino acid k as:

$$D_{ik} = f_{ik} \log \frac{f_{ik}}{q_k} \tag{1}$$

where f_ik is the observed frequency that the amino acid k appears at position i, and q_k is the background probability for that amino acid among all proteins for a given species (Supplementary Table 2). Our mutation score at position i is then the difference in consensus relative entropy (D_ic) and wildtype relative entropy (D_iw):

$$S_i = D_{ic} - D_{iw} \tag{2}$$

The homologous sequences were obtained by BLAST search. Canonical human glucose 6-phosphatase (h*G6PC*; UniProt ID: P35575) served as the query sequence to search the "UniProtKB reference proteomes plus Swiss-Prot" database (from uniprot.org) using the BLASTp algorithm with default parameters (BLOSUM62 similarity matrix, expect threshold 10, gap open penalty 11, gap extension penalty 1, no filter) and 100 max sequences (see Source Data File). The resulting 100 sequences were then realigned to the parental h*G6PC* sequence using the multiple sequence alignment (MSA) software tool MAFFT v.7.407. The MSA was "sliced" to remove gaps in the aligned h*G6PC* sequence. Weblogo v.3.7.1 was used to visualize the consensus amino acids at different positions.

**Preparation of microsomes from mouse livers**. Liver tissues (0.5–1 g) were homogenized in homogenization buffer (0.9% NaCl, 10 μl/mg liver) with Polytron homogenizer at 6000 rpm, followed by addition of microsomal buffer (4 mM NaCl, 2% Glycerol, 12 ul/mg liver). The homogenate was centrifuged at 12,000×g for 20 min at 4 °C, followed by recentrifugation of supernatant at 105,000×g for 1 h at 4 °C. The resulting pellet was resuspended in microsomal buffer (1 ul/mg liver) by passing through 23 G × 1 ½ needles 20 times. Samples were flash frozen in liquid

nitrogen and stored at −80 °C for protein expression and enzymatic activity measurement.

**Protein expression analysis**. hG6Pase-α protein expression levels in cell lysates or liver microsomes were measured by standard immune-blotting procedure, using LI-COR odyssey system. Total protein concentration of cell lysates or liver microsomes were quantified by Pierce® BCA Protein Assay kit (Thermo Scientific). Samples were separated by 4–12 % SDS-PAGE gel and transferred to nitrocellulose membranes by dry blotting system (iBlot2, Invitrogen). Membranes were incubated with anti-hG6Pase-α (HPA052324, Atlas Antibodies) and anti-ERP72 (D70D12, Cell Signaling) followed by incubation with (IR)-labeled goat anti-rabbit secondary antibody (IRDye® 800CW, LI-COR). IR-intensity signals were detected and quantified by Odyssey CLx (LI-COR Biosciences). For quantitative analysis, the expression levels of ERP72 were used as an internal control to normalize that of G6Pase-α.

**G6Pase-α activity assay**. G6Pase-α enzymatic activity was measured by the release of inorganic phosphate from G6P using Taussky–Shorr's method[71]. Briefly, in a round bottom 96-well plate, 40 μl of 200 mM G6P, 100–115 μl of 100 mM BIS-Tris buffer (pH 6.5), and 5–20 μl of either transfected cell lysates or liver microsomes were added and incubated at 37 °C for 30 min. Then, 40 μl of 20% trichloroacetic acid (TCA) solution was added to each well and incubated at room temperature for at least 5 min to quench the reaction. Subsequently, the reaction mixture was centrifuged at 1800×g for 20 min to sediment the precipitated protein and other debris. A portion of supernatant (25–50 μl) was transferred to a new transparent flat-bottom 96-well plate and mixed with 50–75 μl of distilled water and 100 μl of premade Taussky–Shorr color reagent (1% ammonium molybdate, 5% Iron (II) sulfate, and 0.5 M sulfuric acid), followed by incubation at room temperature for 5 min. Color development in reactions was measured by absorbance at 660 nm and the released inorganic phosphate (Pi) was determined based on a series of Pi standards. Final G6Pase-α enzymatic activity was expressed as amount of Pi (nmol) released per mg of total protein per minute of reaction time (nmol/min/mg total protein). The total protein concentration in cell lysates and microsomes was determined by Pierce® BCA Protein Assay kit (Thermo Scientific).

**Confocal immunocytochemistry analysis**. HeLa cells were plated in 96-well, plastic bottom plates (655892, GreinerBio) using recommended culturing conditions, at a density of 15,000 cells per well. Cells were either kept non-transfected or transfected with the h*G6PC* mRNA (50 ng/well) using Lipofectamine 2000. At 6, 24, and 48 h post-transfection, the cells were fixed in 4% PFA, permeabilized in 0.5% Triton X-100, blocked in 1% BSA, followed by immunofluorescent staining with anti- G6Pase-α rabbit Ab (HPA052324, Sigma) and anti-Calnexin mouse Ab (66332, Abcam) or anti-TOM20 mouse Ab (612278, BD BioSciences) to examine the subcellular localization. Secondary antibody incubation was used to amplify the signal (goat anti-rabbit Alexa 488 and goat anti-mouse Alexa 647, respectively). The cells were counter stained with DAPI for nuclei visualization. For image acquisition and colocalization analysis, samples were imaged on the Opera Phenix spinning disk confocal microscope (Perkin Elmer), using a 63 × water immersion objective (NA 1.15). Sixteen fields of view (~40 cells each) have been imaged for each sample. The TOM20 mitochondrial marker was imaged with the 647 nm laser line, and the hG6Pase-α was imaged with the 488 nm laser line and the nuclear stain was imaged with the 405 nm laser line. A z-stack of five optical sections spanning 2 μm were acquired for all three channels. Image analysis was performed in Harmony, using a custom script to calculate the Mander's colocalization coefficient.

**Hepatic h*G6PC* mRNA, G6Pase-α protein, and enzymatic activity analysis in wild-type mice**. WT (CD-1) male mice were i.v. administered with 1.0 mg/kg of eGFP, h*G6PC*-WT, or h*G6PC*-S298C mRNA-LNP and sacrificed at 6 h, 1, 3, 7, or 14 days (*n* = 4). Subsequently, hepatic h*G6PC*-WT and -S298C mRNA levels were measured by RT-qPCR. Briefly, total RNA was extracted from liver tissue using Promega Maxwell RSC simplyRNA tissue kit (Promega A1340) and quantified with Quanti-IT Broad kit (ThermoFisher Scientific Q10213). In total, 10 ng of RNA was used in RT-qPCR reaction using ABI Quant Studio Flex7 instrument and Taqman assay specifically designed to measure hG6PC-specific fragments (forward primer sequence: GTGGCTCCCTTTCAGACTTAG; reverse primer sequence:

GAAGCTCAGCACGTAGAACA; FAM-tagged probe sequence: AAG-GAGGCTTCAGGCTGTCGAAC) (see Supplementary Table 3). hG6PC-WT or -S298C mRNA levels were calculated based on standard curves and normalized to β-actin mRNA level quantified by Taqman assay #Mm02619580_g1 (Thermo-Fisher Scientific). Hepatic hG6Pase-α WT or S298C protein expression and enzymatic activity were measured as described above. Half-lives were determined by non-compartmental analysis using Phoenix WinNonlin (Version 8.1, Certara). For laboratory animals used in Moderna facilities, all experimental protocols were approved by the Institutional Animal Care and Use Committees at Moderna and complied with all relevant ethical regulations regarding the use of research animals. Mice were housed under the following conditions: temperature—68 to 79 °F (20–26 °C), humidity—30–70%, dark/light cycle—an automatically controlled 12-h light:12-h dark—light cycle was maintained.

**Liver-specific knockout (L.G6pc⁻/⁻) mouse model of GSD1a.** The development of a liver-specific G6pc knockout mouse model (L.G6pc⁻/⁻) was previously described[53]. Briefly, a mouse line in the C57BL/6 J background with two loxP sites flanking the G6PC exon 3 (B6.G6PC lox/lox) was generated and crossed with transgenic mice with liver-specific expression of CRE recombinase under the control of the serum albumin promoter fused to a ligand-binding domain of the estrogen receptor (B6.SA CREERT2/w). To induce the excision of G6PC exon 3, the resulting B6.G6PC lox/lox.SAcreERT2/w male adult (6–8 weeks old) mice were injected intraperitoneally once daily with 100 μl of tamoxifen (1 mg/ml, Sigma–Aldrich) for 5 consecutive days, to obtain L.G6pc⁻/⁻ mice. Male mice were housed for a minimum of 4 weeks following the tamoxifen treatment prior to enrolling in the studies. As previously reported, the 4-week period is sufficient to ensure that all mice harbor the gene deletion[53]. Control C57BL/6 J male mice were also treated with similar tamoxifen injections to rule out any potential effect of tamoxifen in treatment outcome. Male mice were housed in the animal facility of Lyon 1 University under temperature controlled (22 °C) conditions and with a 12/12-hour light/dark cycle. Mice had free access to water and standard chow diet. Fasted mice were provided with continuous access to water. All the procedures were performed in accordance with the principles and guidelines established by the European Convention for the Protection of Laboratory Animals. All conditions and experiments were approved by the University Lyon I animal ethics committee and the French Ministry of National Education, Higher Education and Research (Permit Apafis numbers: 20821-2019052414026539v2 and 25143-2020041814543626 v1).

### In vivo efficacy studies

*Dose-ranging study.* L.G6PC⁻/⁻ mice were given a single i.v. bolus injection (via the caudal vein) of either eGFP mRNA (1 mg/kg) or hG6PC-S298C mRNA formulated in LNP (0.2 or 0.5 or 1.0 mg/kg mRNA). Wild-type (C57BL/6 J) mice were treated with PBS. Immediately after the injection, fasting was induced by removal of food and blood glucose levels were measured at 0 (fed state), 2.5, 6, and 24 h of fasting. Mice were euthanized at 24 h post-treatment and livers were harvested, weighed, photographed, and snap-frozen for downstream processing. Hepatic glycogen, G6P, triglycerides, and serum biomarkers including liver enzyme (ALT) and triglycerides were measured by commercially available kits as described below. The G6Pase-α protein expression and G6Pase-α activity in liver microsomes were assessed as described above.

*Duration-of-action study.* Wild-type (C57BL/6 J) male mice or L.G6pc⁻/⁻ male mice were i.v. administered with either PBS or LNP-formulated eGFP mRNA (1.0 mg/kg) or the hG6PC-S298C mRNA (0.5 or 1.0 mg/kg). Subsequently, blood glucose levels were measured over 14 days (on 0, 2, 4, 7, 10, and 14 days post-treatment) with a glucometer (Roche Diagnostic) at 2.5 and 6 h of fasting.

*Repeat-dose efficacy study.* Wild-type (C57BL/6 J) male mice or L.G6pc⁻/⁻ male mice were i.v. administered with five consecutive injections of either eGFP mRNA or hG6PC-S298C mRNA every 10 (second dose) to 14 days (all other doses) at 0.25 mg/kg dose level. After each treatment, blood glucose was measured at 2.5 h of fasting on days 1, 4, 7, and 10.

*Hepatocellular adenoma (HCA) prevention study.* Wild-type (C57BL/6 J) (n = 21) and L.G6pc⁻/⁻ male mice (n = 60) were fed a high fat/high sucrose (HF/HS) diet throughout the course of the study (and 3 months prior to beginning treatment with mRNAs) to accelerate/facilitate the development of HCA/HCCs[54]. Male mice were treated with 8–10 consecutive injections of either PBS, eGFP mRNA, or hG6PC-S298C mRNA administered i.v. every 7 to 14 days at 0.25–0.5 mg/kg dose level. Mice were euthanized 8 days after the last mRNA treatment and livers and tumors were harvested, weighed, counted, measured, and photographed. Liver and tumor tissues were either, snap-frozen in liquid nitrogen and kept at −80°C, or fixed and embedded in paraffin blocks for further use. Total RNA was extracted from liver tissue by the Promega Maxwell RSC simplyRNA tissue kit as mentioned above, and the HCA/HCC-related mRNA markers (β-catenin, transforming growth factor beta-1 and glutamine synthetase) were measured by custom Taqman assays from ThermoFisher Scientific (Supplementrary Table 3). HCA/HCC-related

protein markers (PKM2, β-catenin, and p62 were measured by standard immunoblotting procedure as described above with the following primary antibodies: PKM2 (D78A4) XP® Rabbit mAb (cat #4953, Cell Signaling), β-catenin (D10A8) XP® Rabbit mAb (cat #8480, Cell signaling), and Anti-SQSTM1/p62 mouse mAb (cat #ab56416, Abcam).

### Hepatic biomarker measurements

*Liver G6P measurements.* Hepatic G6P was measured with G6P assay kit (MAK014, Sigma-Aldrich) by following the manufacturer's protocol. Briefly, mouse livers were homogenized in ice-cold H2O (4 μl/mg tissue), followed by centrifugation at 13,000xg, for 10 min at 4 °C. Supernatant was filtered through 10 KDa MWCO spin filter and centrifuged at 13,000xg for 30 min at 4 °C for removal of insoluble materials and proteins. Samples were mixed with reaction mix and incubated for 30 min at room temperature in the dark. Color change in reaction was measured by absorbance at 450 nm (A450). Amount of G6P was determined by subtracting A450 of blanks from that of samples, interpolating values based on a G6P standard curve and normalizing by total protein amount determined by BCA assay.

*Liver glycogen measurements.* Hepatic glycogen was measured with glycogen assay kit (MAK016, Sigma-Aldrich) by following the manufacturer's protocol. Briefly, mouse livers were homogenized in ice-cold H2O (10 μl/mg tissue) and boiled for 5 min. Then, samples were centrifuged at 13,000xg for 5 min at 4 °C for removal of insoluble materials, followed by addition of hydrolysis enzyme mix to samples and glycogen standards. Hydrolysis reaction was carried out for 30 min at room temperature. Subsequently, samples were mixed with reaction mix and incubated for 30 min at room temperature in the dark. Color change in reaction was measured by absorbance at 570 nm (A570). Amount of glycogen was determined by subtracting A570 of blanks from that of samples, interpolating values based on a glycogen standard curve and normalizing by total protein amount determined by BCA assay.

*Liver triglycerides measurements.* Hepatic and serum triglycerides were measured with triglyceride assay kit (10010303, Cayman Chemical) by following the manufacturer's protocol. Briefly, mouse livers were homogenized in NP40 substitute assay reagent (5 μl/mg tissue) and centrifuged at 10,000xg for 10 min at 4 °C. Subsequently, the supernatant was diluted tenfold in NP40 substitute assay reagent and mixed with enzyme mixture. Enzymatic reaction was carried out for 15 min at room temperature. Color change in reaction was measured by absorbance at 545 nm (A545). Amount of triglyceride was determined by subtracting A545 of blanks from that of samples, interpolating values based on a triglyceride standard curve and normalizing by total protein amount determined by BCA assay.

### Safety Studies

*Proinflammatory cytokine analysis.* Proinflammatory cytokine levels were measured from serum of L.G6pc⁻/⁻ mice treated with hG6PC S298C-LNP at 0.5, and 1.0 mg/kg, using a modified MesoScale Diagnostics proinflammatory panel1 mouse kit (MesoScale Diagnostics, #K15048D) with IFNγ, IL-1β, TNFα, and IL-6, only. Briefly, serum samples and cytokine standards were incubated on a plate precoated with anti-cytokine capture antibodies for 2 h at room temperature, followed by washing and incubation with detection antibodies for 2 h at room temperature. Absorbance was read with QuickPlex SQ 120 with the addition of 150 μL 2X MSD Read Buffer. Serum cytokine levels were calculated based on respective standard curves.

*Plasma alanine aminotransferase (ALT) analysis.* Plasma ALT levels were measured by using a commercially available kit (MAK052, Sigma-Aldrich) following the manufacturer's protocol. In this assay, ALT is determined by the amount of pyruvate generated. The ALT activity is expressed as milliunit (mU)/ml, where one mU of ALT is defined as amount of enzyme that generates 1 nmol of pyruvate per minute at 37 °C.

*Anti-G6Pase-α antibody ELISA.* Antibodies against G6Pase-α were quantified on Nunc Immuno Maxisorp plates (ThermoFisher, #442404) coated with 0.5 μg/mL recombinant G6Pase-α protein (Viva Biotech) in 50 mM Na2CO3 for 1 h at room temperature and blocked with SuperBlock (PBS) Blocking Buffer (ThermoFisher, #37515). Mouse serum diluted 1:20 dilution in PBS was incubated 1 h at room temperature and quantified with a standard curve using commercial rabbit anti-human G6Pase-α IgG (Abcam, #ab93857) at 0–2 μg/mL. Samples and standards were incubated 1 h at room temperature with goat anti-rabbit IgG-HRP (Abcam, #ab6721) or goat anti-mouse IgG H + L-HRP (Fitzgerald Laboratories, #43-GM30) secondary antibody at 1:100,000 dilution. ELISA was developed with 1-Step Ultra TMB-ELISA substrate (ThermoFisher, #34028) and Stop Solution (ThermoFisher, #SS04) before reading at 450 nm.

**Statistical analysis.** All data were shown as means ± SD. For statistical analysis, unless otherwise stated, raw data were Log2 transformed to account for non-Gaussian distribution and means were compared by one-way analysis of variance

(ANOVA), followed by Dunnett's post hoc test for multiple comparisons, using GraphPad Prism v7 software (GraphPad Software). For the duration-of-action study, a two-sample $t$-test (two-sided) was used to compare the blood glucose level of eGFP mRNA treated group to that of h$G6PC$ S298C mRNA treated groups over 14 days. The multiple testing was corrected by Bonferroni adjusted level of 0.005. Repeat dose study was also analyzed with a two-sample $t$-test (two-sided), comparing the blood glucose level of eGFP mRNA treated group to that of h$G6PC$ S298C mRNA treated groups over 52 days. The multiple testing was also corrected by Bonferroni adjusted level of 0.005.

**Reporting Summary**. Further information on research design is available in the Nature Research Reporting Summary linked to this article.

## Data availability

The authors declare that all relevant data supporting the findings of this study are available within the article and its Supplementary Information files. Source data are provided with this paper.

## Code availability

The protein sequence alignment was performed using MAFFT v.7.407 (https://mafft.cbrc.jp/alignment/software/). The consensus amino acids at different positions were visualized using Weblogo v.3.7.1 (https://github.com/WebLogo/weblogo). A Jupyter notebook containing Python code to identify and rank the consensus substitutions is available from the corresponding authors upon request.

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

## Acknowledgements

The authors acknowledge the contribution of Moderna's Molecular Biology, Protein Sciences and preclinical production groups. The authors would like to thank Christine Lukacs, Lin Guey, David Reid, Nicholas Amato, David Marquardt for helpful discussions, and Simone Mori for valuable clinical discussions and for reviewing the manuscript. The authors also thank the members of Animalerie Lyon Est Conventionnelle et SPF (ALECS, SFR Santé Lyon-Est, Université Claude Bernard Lyon 1, Lyon, France) for animal housing and care; and Cécile Saint Béat, Clara Bron and Carine Zitoun (INSERM UMR1213, Lyon, France) for monitoring animal welfare.

## Author contributions

Conceptualization: F.R., P.F.F., P.G.V.M., and P.H.G.; Methodology: P.G.V.M., P.H.G., F.R., A.F., G.B., C.T., J.S., C.M., K.B., A.-R.G., A.D., V.P., C.P., W.Z., L.C., and M.H.; Data acquisition, curation, and analysis: J.C., M.C., E.G., M.Si., M.So., V.V., E.W., A.M., J.Z., S.L., L.Y., A.F., A.-R.G., T.K., Z.Z., B.L., M.W., B.T., E.O., A.D., V.P., C.P., W.Z., L.C., L.R., V.N., M.Z., and U.R.; Writing—original draft: J.C., M.C., A.-R.G., A.D., V.P., G.B., W.Z., F.R., and P.H.G.; Writing— review and editing: J.C., M.C., P.H.G., E.M., F.R., and G.M.; Supervision: P.G.V.M., F.R., and P.H.G.

## Competing interests

J.C., M.C., E.G., E.D., J.Z., S.L., L.Y., A.F., A.-R.G., K.B., T.K., C.M., Z.Z., B.L., G.B., M.W., B.T., C.T., E.O., J.S., A.D., V.P., C.P., W.Z., L.C., M.H., E.M., L.R., V.N., M.Z., U.R., P.F.F., P.G.V.M., and P.H.G. are employees of and receive salary and stock options from Moderna Inc. M.Si., M.So., V.V., E.W., A.M., G.M., and F.R. declare no competing interests.
