## [Peer Review File · Nature Communications]

Reviewers' Comments:

Reviewer #1:

Remarks to the Author:

This is a well written manuscript with clear and compelling results. Although this article is more specialized to mRNA therapeutics, there are some elements that may make it of interest to a broader multidisciplinary audience.

1. Figure 1b, 1c, 1d, and 6d, the graph lettering / spacing did not transfer correctly to PDF from the original graph. This can be corrected.
2. Please check and verify all statistical comparisons.
3. Please add the molar ratios of lipids used in the LNP.

Reviewer #2:

Remarks to the Author:

The manuscript from Cao et al. presents a very detailed analysis of an improved mRNA therapy for GSD1a. The study was well-designed and the data are extensive. Statistical analyses are appropriate. Given the previously published data in Reference 41 and the limited benefits demonstrated here, the novelty of the approach and potential impact in the field are moderate. Concerns to be addressed are listed below and prioritized as major or minor.

Major:

- 1) Characterization of GSD1a and alternative therapies should be more accurate and objective.
 - a. Introduction, page 3, paragraph 1:
 - i. Not all GSDs are "characterized by failure to maintain euglycemia".
 - ii. GSD1a is not the "most severe form of hepatic GSDs". GSD IV is lethal in childhood.
 - b. Introduction, page 4 bottom to page 5 top: It is not true that it is not possible for gene therapy to avoid "the risk of constitutive activation of protein function"... "something not possible with gene therapy." References 19 and 21 use a regulated promoter that only produces G6Pase-alpha as needed to prevent hypoglycemia with an AAV vector, which is regulated expression and prevents the toxicity for constitutive activation of protein function. On the other hand, the risk for excessive G6Pase-alpha is present for mRNA and this should be acknowledged in this manuscript.
- 2) The benefits of mRNA therapy are transient and limited. As shown in Figure 5, statistically significant improvement is only seen for up 4 days following mRNA treatment. How is this an adequate treatment for a congenital disease? How would this be administered to prevent complications clinically, a twice weekly infusion? This does not seem to be an effective treatment, and it would be very costly and inconvenient. These limitations should be clearly stated in the Discussion. Furthermore in the Discussion (page 13, first paragraph) it is falsely stated that repeat mRNA treatment "may reduce the risk of long-term complications (eg HCA/HCC), which is not supported by the data.
- 3) The safety evaluation is incomplete. The abstract claims that safety is demonstrated, which should be qualified as "supported", not demonstrated. Specifically, there is no data regarding food consumption, pathologic lesions or other post-mortem abnormalities, mortality, observations of hypersensitivity (body temperature, altered breathing following administration), or T cell infiltrates in the tissues. This is not a preclinical study that completely investigate safety, at least not without these details listed above.

Minor:

- 1) Figure 1b, 1c, and 3a-c : labels and legends are so small that they are illegible, preventing easy interpretation of the data.
- 2) Figure 3: The baseline values for mice prior to treatment are not shown for comparison.
- 3) Fasting interval in long-term experiment is very brief at 2.5 hours. This should be clearly stated in Results.

4) The number of males and females for every experiment should be specified in the figure legends. Sex is an important biological variable that influences the response to fasting and to treatment.

5) Page 12, paragraph 1: States that AFP and CRP were normal, but this data is not shown in Figure 6.

Reviewer #3:

Remarks to the Author:

The study by Cao and coworkers (NCOMMS-20-33589-T) examined the efficacy of mRNA-based therapy for GSD-Ia using L-G6pc^{-/-} mice lacking hepatic G6Pase- α activity. The current work is an extension of earlier studies by Roseman et al (Mol Ther 26:814-821, 2018) showing that a single administration of LNP-encapsulated G6PC mRNA corrected fasting hypoglycemia and normalized metabolic abnormalities in L-G6pc^{-/-} mice. Roseman et al also reported two engineered protein variants of human G6Pase- α with increased duration of expression and showed that these G6PC variants displayed increased efficacy and G6Pase- α activity was detectable up to 12 days after a single application. The new information reported in the present study is the demonstration that repeated dosing of LNP-G6PC mRNA is safe which is critical for a chronic disorder like GSD-Ia.

A significant concern with this manuscript is the lack of acknowledgement of earlier work in the field. The authors present the search for a hG6Pase- α enzyme with increased activity and in particular the discovery of hG6PC-S298C and co-hG6PC-S298C variants as new findings. This is inaccurate. The search for enzymes of increased activity and the potential therapeutic significance of the hG6PC-S298C and co-hG6PC-S298C variants in particular, their use and advantages in gene therapy have already been reported (Zhang et al J Inherit Metab Dis 42:470-479, 2019; Zhang et al Biochem Biophys Res Commun 527:824-830, 2020), and are published in a US patent (#10,415,044). The rAAV8-co-hG6PC-WT vector (Kim et al Mol Genet Metab 120: 229-234, 2017) is currently in phase I/II clinical trials for human GSD-Ia (NCT03517085). Although codon-optimization is now a common procedure used to increase the activity/stability of a transgene, Cao et al should discuss this since the co-hG6PC-S298C mRNA used in the present study is a codon-optimized hG6PC variant.

One unique contribution of the study is the generation of the co-hG6PC-S298C variant with increased enzymatic activity and that treatment of L-G6pc^{-/-} mice by LNP-co-hG6PC-S298C mRNA normalized hypoglycemia and metabolic abnormalities of the mice. Unfortunately, Gao et al were unable to conduct literature search to know that these are not new findings. In this context, the primary original finding of the present study is only the demonstration that repeated dosing of LNP-co-hG6PC-S298C mRNA is safe, like many other reported mRNA therapy studies.

Specific Comments

1. The title of this study overstated their results. The authors have conducted experiments on the effects of LNP-co-hG6PC S298C mRNA on hepatic tumor prevention, but the data are not statistically significant. In order to reach a statistically significant conclusion, the authors should conduct additional experimentation.

2. The L-G6pc^{-/-} mice used in this study manifest a phenotype mimicking that of human GSD-Ia except the mice exhibit normoglycemia in the fed state and can survive prolonged fasting (45 h) (Mutel et al Diabetes 60:3121-31, 2011) in contrast to human GSD-Ia patients and the global G6pc^{-/-} mice. Indeed, induction of glucose production from the kidney and intestine has rescued the L-G6pc^{-/-} mice from suffering hypoglycemic seizures, the hallmark of human GSD-Ia. Although measuring the 6-hour-fasted blood glucose levels in L-G6pc^{-/-} mice appears adequate to monitor correction of fasting hypoglycemia, the physiological relevance to human GSD-Ia is not clear. It would be better to conduct a few experiments using the global G6pc^{-/-} mice that suffer

from severe fasting hypoglycemia mimicking human GSD-Ia.

3. Gao et al examined the relative efficiency of hG6PC-WT, co-hG6PC-WT, hG6PC-S298C, and co-hG6PC-S298C mRNAs in CD-1 mice that are not deficient in G6Pase- α (Figure 2a and 2b). The reviewer failed to understand why use CD1 mice instead of the L-G6pc^{-/-} mice for this important study. Roseman et al (Mol Ther 26:814-821, 2018) compared the efficacy of hG6PC-WT, hG6PC-V1, and hG6PC-V2 mRNAs and showed that all three mRNAs achieved near-normal blood glucose levels but only hG6PC-V1, and hG6PC-V2 mRNAs can last 12 days. Gao et al should conduct efficacy study of hG6PC-WT, co-hG6PC-WT, hG6PC-S298C, and co-hG6PC-S298C mRNAs using the L-G6pc^{-/-} mice.

4. The L-G6pc^{-/-} mice were generated by tamoxifen-mediated excision of exon 3 from the mouse G6pc gene in the liver. Studies have shown that complete deletion of hepatic G6Pase- α activity will take 7-10 days post-tamoxifen treatment. The authors should state clearly in Methods how many days post-tamoxifen treatment to conduct the mRNA therapy. The same question also applies to their study on the effect of co-hG6PC-S298C mRNA on prevention of HCA/HCC.

5. The authors used an antibody (HPA052324, Atlas Antibodies) to measure the anti-hG6Pase- α antibodies in the sera of co-hG6PC-S298C mRNA-treated L-G6pc^{-/-} mice. Please present the actual Western blots for this analysis to ensure the antibody used is specific for human G6Pase- α . The authors should also include a positive and a negative control in Western-blot analysis.

6. Figure 6: the authors indicated that serum biomarkers associated with GSD1a (glycemia) and HCA/HCC (AFP and CRP) trended towards normal levels upon treatment with hG6PC S298C mRNA (Fig. 6e-f). Where are the data for AFP and CRP?

Rebuttal

We thank the reviewers for their detailed and helpful comments on our manuscript. We have addressed each of the comments and revised the manuscript accordingly. Changes in the manuscript text are highlighted in red. Excerpts from the reviewers' critiques are indicated below in *italics*.

Reviewer #1 (Remarks to the Author):

This is a well written manuscript with clear and compelling results. Although this article is more specialized to mRNA therapeutics, there are some elements that may make it of interest to a broader multidisciplinary audience.

1. Figure 1b, 1c, 1d, and 6d, the graph lettering / spacing did not transfer correctly to PDF from the original graph. This can be corrected.

Done

2. Please check and verify all statistical comparisons.

Done

3. Please add the molar ratios of lipids used in the LNP.

Done

Reviewer #2 (Remarks to the Author):

The manuscript from Cao et al. presents a very detailed analysis of an improved mRNA therapy for GSD1a. The study was well-designed and the data are extensive. Statistical analyses are appropriate. Given the previously published data in Reference 41 and the limited benefits demonstrated here, the novelty of the approach and potential impact in the field are moderate. Concerns to be addressed are listed below and prioritized as major or minor.

Major:

1) Characterization of GSD1a and alternative therapies should be more accurate and objective.

a. Introduction, page 3, paragraph 1:

i. Not all GSDs are "characterized by failure to maintain euglycemia".

ii. GSD1a is not the "most severe form of hepatic GSDs". GSD IV is lethal in childhood.

We agree with the reviewer and have revised the manuscript text to clarify this point (see revised Introduction section, page 3).

b. Introduction, page 4 bottom to page 5 top: It is not true that it is not possible for gene therapy to avoid "the risk of constitutive activation of protein function"... "something not possible with gene therapy." References 19 and 21 use a regulated promoter that only produces G6Pase-alpha as needed to prevent hypoglycemia with an AAV vector, which is regulated expression and prevents the toxicity for constitutive activation of protein function. On the other hand, the risk for excessive G6Pase-alpha is present for mRNA and this should be acknowledged in this manuscript.

We agree with the reviewer and have revised the manuscript text to clarify and down-play this point (see revised Introduction section, page 4-5). Furthermore, the comment regarding constitutive and/or prolonged activation of protein function in the Introduction section is generalized to include those gene

[Type here]

therapies currently approved in humans which, are not under a tissue specific promoter. We also, clarify that the linear dose response observed with mRNA therapy may allow titrating an ideal dose for each patient, thereby mitigating the risk of excessive G6Pase- α expression. To this end, we have updated the text and references accordingly. We also provide a comparison of our mRNA therapy with Ultragenyx's GSD1a gene therapy (in PhI/II) in the Discussion section, page 14.

2) The benefits of mRNA therapy are transient and limited. As shown in Figure 5, statistically significant improvement is only seen for up 4 days following mRNA treatment. How is this an adequate treatment for a congenital disease? How would this be administered to prevent complications clinically, a twice weekly infusion? This does not seem to be an effective treatment, and it would be very costly and inconvenient. These limitations should be clearly stated in the Discussion.

We agree that dosing every four days may be impractical for clinical application. That said, based on our duration of action and repeat dose studies our estimated dosing regimen/frequency of 7 days (weekly dosing) not 4 days was sufficient to maintain euglycemia (indicated by therapeutic threshold of 60mg/dL or above) in this mouse model. In addition, our expectation is that the necessary dosing frequency will be substantially lower, likely at 3-week intervals or longer, due to slower drug metabolism in humans vs. mice. This is based on the following:

1) Many studies of diverse therapeutic classes including small molecule therapies and alternative nucleic acid therapeutic categories that have been shown to require substantially less frequent dosing in humans vs. mice (we now mention several of these studies in the discussion for context).

2) Analysis derived from a novel unpublished (Moderna) human PK/PD model that was trained with data from similar mRNA drugs to extrapolate from rodent and/or NHP pharmacology studies to identify appropriate dosage regimens for humans. (We summarize the relevant analysis below, but are not able to include it in the manuscript as Moderna is preparing a separate manuscript for publication that describes the model in detail.) Preliminary evaluation of simulated relationship between hG6PC S298C mRNA dose and hG6Pase protein expression using the scaled human PK/PD model indicates that ≥ 0.1 mg/kg dose of the hG6PC S298C mRNA is predicted to yield a hepatic hG6Pase- α S298C concentration of ≥ 5 μ g/g (predicted to be enough to achieve therapeutic threshold – maintenance of blood glucose levels above 60 mg/dL) for at least 3 weeks post a single dose (our unpublished data). Thus, assuming the relationship between hG6Pase- α S298C levels and fasting blood glucose concentrations is similar between L.G6pc-/- mice and GSD1a patients, the hG6PC S298C mRNA dose yielding ≥ 5 μ g/g dosed every 3 weeks is predicted to be efficacious in humans, and considered a viable dosing regimen for GSD1a patients.

Furthermore in the Discussion (page 13, first paragraph) it is falsely stated that repeat mRNA treatment “may reduce the risk of long-term complications (eg HCA/HCC), which is not supported by the data.

We now provide additional robust evidence that hG6PC S298C mRNA therapy is effective in preventing hepatic adenomas and carcinomas in the liver-specific mouse model of GSD1a (see new Fig 6 and Supplementary Fig. 4).

3) The safety evaluation is incomplete. The abstract claims that safety is demonstrated, which should be qualified as “supported”, not demonstrated. Specifically, there is no data regarding food consumption, pathologic lesions or other post-mortem abnormalities, mortality, observations of hypersensitivity (body temperature, altered breathing following administration), or T cell infiltrates in the tissues. This is not a

[Type here]

preclinical study that completely investigate safety, at least not without these details listed above.

While we agree with the reviewer that additional safety studies are warranted prior to testing in humans, our initial report of safety is primarily intended to assess potential immune effects due to mRNA therapy (inflammatory response) and introducing a 'foreign' protein into a naïve model (assessed by anti-drug antibody assay). That said, we did not observe hypersensitivity (changes in body temperature, altered breathing, ruffled fur), mortality, changes in behavior (i.e. loss of appetite, distress) in treated animal (this is now stated in the manuscript). Furthermore, we have not observed T cell infiltrates upon histological evaluation of tissues in several studies from our group using similar mRNAs (harboring same chemistry to the one described in this study) and LNPs (An et al Cell Rep. 2018; Jiang et al Nat Med. 2018; An et al, EBioMedicine 2019; Balakrishnan et al Mol Ther 2020; Jiang et al Nature Comm. 2020; Sabnis et al, Mol Ther 2018).

We have now modified the text in the Results section on page 11 to indicate that while additional safety studies performed in larger animal models (i.e. rats and nonhuman primates) are warranted for future clinical development, the above data suggest that hG6PC S298C mRNA may be well-tolerated under the conditions of these studies.

Minor:

1) Figure 1b, 1c, and 3a-c: labels and legends are so small that they are illegible, preventing easy interpretation of the data.

We have enlarged fonts and labels to make these legible.

2) Figure 3: The baseline values for mice prior to treatment are not shown for comparison.

We now clearly indicate that the eGFP control group serves as baseline for the data presented in Fig. 3.

3) Fasting interval in long-term experiment is very brief at 2.5 hours. This should be clearly stated in Results.

Done (see comment on page 10 of results section).

Based on the single-dose efficacy studies (**Fig. 4a and 5a**), the 2.5-hr post-fasting blood glucose level was determined to be equally predictive of efficacy as the longer 6-hr post-fasting glucose level.

Furthermore, it has been reported that a 2.5-hr fast in mice is equivalent to a 6 to 8-hr fast in humans, while a 5-6-hr fast is comparable to a 16 to 18-hr fast in humans⁵⁴. To more closely mimic an overnight fast in humans, in the multi-dose study (**Fig. 5b**), we monitored blood glucose at 2.5-hr post-fasting on days 0, 1, 4, 7, and 10 following administration of the mRNA.

4) The number of males and females for every experiment should be specified in the figure legends. Sex is an important biological variable that influences the response to fasting and to treatment.

Done

5) Page 12, paragraph 1: States that AFP and CRP were normal, but this data is not shown in Figure 6.

We provide additional data to support our claim that mRNA therapy significantly prevents formation of HCA/HCCs in the mouse model. We have revised this statement and data accordingly. (see new Fig. 6 and Supplementary Fig. 4).

Reviewer #3 (Remarks to the Author):

The study by Cao and coworkers (NCOMMS-20-33589-T) examined the efficacy of mRNA-based therapy

[Type here]

for GSD-1a using L-G6pc^{-/-} mice lacking hepatic G6Pase- α activity. The current work is an extension of earlier studies by Roseman et al (Mol Ther 26:814-821, 2018) showing that a single administration of LNP-encapsulated G6PC mRNA corrected fasting hypoglycemia and normalized metabolic abnormalities in L-G6pc^{-/-} mice. Roseman et al also reported two engineered protein variants of human G6Pase- α with increased duration of expression and showed that these G6PC variants displayed increased efficacy and G6Pase- α activity was detectable up to 12 days after a single application. The new information reported in the present study is the demonstration that repeated dosing of LNP-G6PC mRNA is safe which is critical for a chronic disorder like GSD-1a.

A significant concern with this manuscript is the lack of acknowledgement of earlier work in the field. The authors present the search for a hG6Pase- α enzyme with increased activity and in particular the discovery of hG6PC-S298C and co-hG6PC-S298C variants as new findings. This is inaccurate. The search for enzymes of increased activity and the potential therapeutic significance of the hG6PC-S298C and co-hG6PC-S298C variants in particular, their use and advantages in gene therapy have already been reported (Zhang et al J Inherit Metab Dis 42:470-479, 2019; Zhang et al Biochem Biophys Res Commun 527:824-830, 2020), and are published in a US patent (#10,415,044).

We apologize to the reviewer if we gave the impression that we were ignoring previous studies. Indeed, it was never our intention to claim that we were the first to describe the S298C protein variant. We are well-aware of the work by Zhang et al. and have modified the Introduction section to more clearly emphasize their contributions. See revised Introduction, page 6.

What we describe in the manuscript is a bioinformatics approach based on consensus protein sequence analysis of mammalian G6Pase orthologs that led us to the identification of the S298C variant as previously reported by Zhang et al. Thus, our work represents an independent discovery of the protein variant hG6PC_S298C (along with other variants with increased expression and activity). Furthermore, combination of the protein engineering approach with the codon-optimization modification at nucleotide level makes this study a unique contribution to the field (as recognized by the reviewer) mainly because this represents the first demonstration of the strength of mRNA-based replacement therapies for genetic disease – we can come up with a much improved version of protein expressed in a subcellular domain (ER membrane) that was otherwise intractable with traditional enzyme/protein replacement therapy.

The rAAV8-co-hG6PC-WT vector (Kim et al Mol Genet Metab 120: 229–234, 2017) is currently in phase I/II clinical trials for human GSD-1a (NCT03517085). Although codon-optimization is now a common procedure used to increase the activity/stability of a transgene, Cao et al should discuss this since the co-hG6PC-S298C mRNA used in the present study is a codon-optimized hG6PC variant.

Done.

One unique contribution of the study is the generation of the co-hG6PC-S298C variant with increased enzymatic activity and that treatment of L-G6pc^{-/-} mice by LNP-co-hG6PC-S298C mRNA normalized hypoglycemia and metabolic abnormalities of the mice. Unfortunately, Gao et al were unable to conduct literature search to know that these are not new findings. In this context, the primary original finding of the present study is only the demonstration that repeated dosing of LNP-co-hG6PC-S298C mRNA is safe, like many other reported mRNA therapy studies.

Also, we have revised the manuscript to emphasize (and clarify) that the novelty of our work is the demonstration of long-term safety and efficacy, following repeat dosing, of an mRNA therapy in model mice that manifest with underlying liver pathology (i.e., potentially precluding efficient and sustained mRNA delivery to liver). Indeed, previous studies from our group evaluating long-term safety of our

[Type here]

mRNA therapy for other liver metabolic diseases, have been performed in model mice with minimal to no liver pathology (An et al Cell Rep. 2018; Jiang et al Nat Med. 2018; An et al. EBioMedicine 2019; Balakrishnan et al Mol Ther 2020; Jiang et al Nature Comm. In press). In contrast, the mouse model described in this study exhibits hepatic pathologies observed in GSD1a patients, including hepatomegaly, fat accumulation or steatosis, and glycogen overload. Thus, based on the repeat-dosing data reported in this study, this is the first demonstration that an mRNA-LNP therapy can effectively treat liver metabolic diseases which manifest with underlying liver pathology.

Specific Comments

1. The title of this study overstated their results. The authors have conducted experiments on the effects of LNP-co-hG6PC S298C mRNA on hepatic tumor prevention, but the data are not statistically significant. In order to reach a statistically significant conclusion, the authors should conduct additional experimentation.

See Reviewer 2, comment 2 above. We now provide additional robust evidence that Moderna's mRNA therapy is effective in preventing hepatic adenomas and carcinomas in the liver-specific mouse model of GSD1a (see new Fig 6 and Supplementary Fig. 4).

2. The L-G6pc^{-/-} mice used in this study manifest a phenotype mimicking that of human GSD-1a except the mice exhibit normoglycemia in the fed state and can survive prolonged fasting (45 h) (Mutel et al Diabetes 60:3121-31, 2011) in contrast to human GSD-1a patients and the global G6pc^{-/-} mice. Indeed, induction of glucose production from the kidney and intestine has rescued the L-G6pc^{-/-} mice from suffering hypoglycemic seizures, the hallmark of human GSD-1a. Although measuring the 6-hour-fasted blood glucose levels in L-G6pc^{-/-} mice appears adequate to monitor correction of fasting hypoglycemia, the physiological relevance to human GSD-1a is not clear. It would be better to conduct a few experiments using the global G6pc^{-/-} mice that suffer from severe fasting hypoglycemia mimicking human GSD-1a.

We thank the reviewer for the thoughtful comments and feedback. With regards to the mouse model, we acknowledge that several mouse models of GSD1a exist which range from complete knockout to tissue specific knockout. GSD1a disease severity in these models correlates with extent of G6Pase deficiency. Careful consideration was given to the utility of each murine model for the development of mRNA therapy for GSD1a. The global knockout mouse model recapitulates most of the pathophysiology of human Gsd1a disease, including severe hypoglycemia, hyperlipidemia, hyperuricaemia, growth retardation, hepatomegaly, and nephromegaly. The model mouse also displays age-dependent glycogen accumulation in both liver and kidney, and severe hepatic steatosis. However, most of these mice do not survive beyond 5 weeks of age, despite extensive nutritional care including frequent injections of glucose, making it difficult to maintain the model mouse to investigate the long-term complications and use it as a tool to evaluate the therapeutic potential of a GSD1a treatment. To overcome these challenges, a liver-specific knockout mouse model of GSD1a has been developed. This mouse model was developed by targeted deletion of the G6Pase- α -encoding *G6pc* gene in liver (*L.G6pc^{-/-}*) through an inducible CRE/LoxP strategy by our collaborator, Dr. Fabienne Rajas and was the model we favored to assess the long-term effects of our mRNA therapy for the treatment of GSD1a. Like GSD1a patients, the *L.G6pc^{-/-}* mice are unable to mobilize hepatic glycogen reserves and convert glycogen derived G6P into glucose leading to severe hypoglycemia and glycogen overload during a fasting state. The *L.G6pc^{-/-}* mice also exhibit hepatic pathologies observed in Gsd1a patients, including hepatomegaly, fat accumulation or steatosis, and glycogen overload. Loss of hepatic G6Pase in these mice results in hepatic accumulation of glucose-6 phosphate (G6P), glycogen, and triglycerides, which, results in hepatomegaly and steatosis. In addition, the *L.G6pc^{-/-}* mice also display dysregulations of plasma lipid and other biochemistry profiles, including increased triglycerides and cholesterol. However, this model does not

[Type here]

exhibit some non-liver complications as seen in GSD1a patients. For instance, chronic kidney pathologies and complications are not represented in this model or within the scope of our study.

Nevertheless, based on the overall phenotypes of this model and our strategies in developing liver-directed mRNA based therapies, this model is considered a relevant GSD1a model as a preclinical tool for evaluating the pharmacological efficacy of mRNA therapy for GSD1a.

Importantly, for the pharmacological intervening studies, mRNA therapy was administered to mice that were at least 3-month old or at least one month after the total deletion of the liver *G6pc*.

3. Cao et al examined the relative efficiency of hG6PC-WT, co-hG6PC-WT, hG6PC-S298C, and co-hG6PC-S298C mRNAs in CD-1 mice that are not deficient in G6Pase- α (Figure 2a and 2b). The reviewer failed to understand why use CD1 mice instead of the L-G6pc-/- mice for this important study. Roseman et al (Mol Ther 26:814-821, 2018) compared the efficacy of hG6PC-WT, hG6PC-V1, and hG6PC-V2 mRNAs and showed that all three mRNAs achieved near-normal blood glucose levels but only hG6PC-V1, and hG6PC-V2 mRNAs can last 12 days. Gao et al should conduct efficacy study of hG6PC-WT, co-hG6PC-WT, hG6PC-S298C, and co-hG6PC-S298C mRNAs using the L-G6pc-/- mice.

While we agree with the reviewer that the L.G6pc-/- mouse model is preferred to WT mice for assessing efficacy of our mRNA therapy, the main purpose for this study was to simply perform a rapid assessment of hepatic protein expression and activity of the codon-optimized variants in mice to facilitate the identification of the optimized construct. Indeed, based on previous experience in performing these types of studies, we have determined the use of WT mice to be the most effective way to rapidly screen through multiple mRNA constructs in vivo. Once the top mRNA constructs with the highest expression and activity were identified, the mRNAs were extensively evaluated for expression, activity, and efficacy in the L.G6pc-/- mouse model.

4. The L-G6pc-/- mice were generated by tamoxifen-mediated excision of exon 3 from the mouse G6pc gene in the liver. Studies have shown that complete deletion of hepatic G6Pase- α activity will take 7-10 days post-tamoxifen treatment. The authors should state clearly in Methods how many days post-tamoxifen treatment to conduct the mRNA therapy. The same question also applies to their study on the effect of co-hG6PC-S298C mRNA on prevention of HCA/HCC.

Done. See revised Methods section page 22.

5. The authors used an antibody (HPA052324, Atlas Antibodies) to measure the anti-hG6Pase- α antibodies in the sera of co-hG6PC-S298C mRNA-treated L-G6pc-/- mice. Please present the actual Western blots for this analysis to ensure the antibody used is specific for human G6Pase- α . The authors should also include a positive and a negative control in Western-blot analysis.

Done. See new Supplementary Fig. 2 in Supplementary Materials.

We also want to clarify that the anti-drug antibody assay performed in Fig. 5f was an ELISA based assay, using (Abcam, #ab93857) as standards for quantifying anti-human G6Pase- α antibodies from sera. This assay was used to measure a potential immune response (antibody) against our target protein (human G6pase α). (HPA052324, Atlas Antibodies) was used for immunoblot analysis and we included actual images of blots to demonstrate the specificity for human G6Pase (see new Supplementary Fig. 2).

6. Figure 6: the authors indicated that serum biomarkers associated with GSD1a (glycemia) and HCA/HCC (AFP and CRP) trended towards normal levels upon treatment with hG6PC S298C mRNA (Fig. 6e-f). Where are the data for AFP and CRP?

[Type here]

See response to reviewer 3, comment 1 above. We now provide additional robust evidence that Moderna's mRNA therapy is effective in preventing hepatic adenomas and carcinomas in the liver-specific mouse model of GSD1a. (see new Fig 6 and Supplementary Fig. 4).

[Type here]

Reviewers' Comments:

Reviewer #1:

Remarks to the Author:

Comments were addressed. I recommend to accept.

Reviewer #2:

Remarks to the Author:

My concerns were well-addressed in the revised manuscript as summarized in the response to reviewers cover letter.

Reviewer #3:

Remarks to the Author:

This is a well-designed study of LNP-G6PC mRNA therapy for GSD-Ia. It builds on the substantial contributions of Roseman et al (Mol Ther 26:814-821, 2018) who have shown that that a single administration of LNP-G6PC mRNA corrects fasting hypoglycemia and normalized metabolic abnormalities in L-G6pc^{-/-} mice, validating the therapeutic approach. The additional findings contributed by the present manuscript is to show that repeated dosing of LNP-G6PC mRNA is safe. In the revision, the authors further showed that repeated dosing of LNP-G6PC mRNA reduced tumor incidence, a new but an expected finding. Previously, Cho et al had shown that treating L-G6pc^{-/-} mice at the tumor-developing stage with rAAV-G6PC restored hepatic G6Pase- α expression, normalized glucose homeostasis and prevented de novo HCA/HCC development (J Inherit Metab Dis 42:459-469, 2019).

A remaining concern about the revised manuscript is the failure to acknowledge earlier work in the field, leaving the implication that the S298C variant and codon optimization they report are novel findings. Zhang et al (J Inherit Metab Dis 42:470-479, 2019) documented in vitro and in vivo characterization of four hG6PC constructs, hG6PC-wild-type, codon optimized (co) hG6PC, hG6PC-S298C, and co-hG6PC-S298C, and showed that the co-hG6PC-S298C is 3 to 4-fold more active than the hG6PC-wild-type construct. Therefore, the statement in the revised manuscript on page 7, paragraph 2 and the statement transmitted to the reviewer, "Furthermore, combination of the protein engineering approach with the codon-optimization modification at nucleotide level makes this study a unique contribution to the field" is misleading. The authors' statements in the revised manuscript on page 6, lines 12-13 and on page 7, lines 3-4 also contradicts the references they do make about the study by Zhang et al. In Results on page 6, lines 12-13, the authors stated, "This finding is consistent with previous studies by Zhang et al., which also report improvements in protein expression with the S298C substitution^{47,48}."; and on page 7, lines 3-4 the authors stated, "In addition to protein engineering, we performed codon optimization of the hG6PC mRNA sequence to further enhance protein expression and enzyme activity."

Specific Comments

1. Page 10, lines 14-16: the statement, "it has been reported that a 2.5-hr fast in mice is equivalent to a 6 to 8-hr fast in humans, while a 5-6-hr fast is comparable to a 16 to 18-hr fast in humans⁵⁸" is misleading. Jensen et al (reference 58) only stated that "Fasting for 5–6 h instead of overnight (16–18 h) might offer a better comparison to humans, who are usually fasted overnight before similar tests." While the L-G6pc^{-/-} mice manifest most clinical manifestations of human GSD-Ia, they exhibit normoglycemia in the fed state and can survive prolonged fasting because the induction of glucose production from the kidney and intestine has rescued the L-G6pc^{-/-} mice from suffering hypoglycemic seizures, the hallmark of human GSD-Ia. The authors should take precaution to extrapolate fasting data obtained using L-G6pc^{-/-} mice to human GSD-Ia patients.

2. Page 15, lines 9-11: the statement, "While encouraging, the gene therapy approach was not able to abrogate pre-existing tumors due to lack of expression of the virus in the adenoma lesions⁶⁸." Reference 68 was incorrectly cited. The reviewer believes Cao et al were referencing the study by Cho et al. (J Inherit Metab Dis 42:459-469, 2019) showing that treating L-G6pc^{-/-} mice at the tumor-developing stage with rAAV-G6PC restored hepatic G6Pase- α expression, normalized glucose homeostasis, and prevented de novo HCA/HCC development. Cho et al further showed that gene therapy could not restore G6Pase- α expression in the HCA/HCC lesions and failed to abrogate the pre-existing tumors. The conclusion was reached via examination of G6Pase- α expression in the tumor by immunohistochemical analysis. Unless Cao et al examined G6Pase- α activity in the tumor of LNP-G6PC mRNA-treated L-G6pc^{-/-} mice by immunohistochemical analysis, the authors cannot conclude, "The expectation is that GSD1a mRNA therapy may, not only prevent de novo HCA/HCC development at the tumor developing stage, but also potentially reduce any pre-existing tumor burden."

Rebuttal

We thank the reviewers for their comments on our revised manuscript. We are encouraged by the positive feedback from this round of revision, and particularly pleased to learn that all comments/concerns from Reviewer 1 and 2 were sufficiently addressed. We have addressed the remaining concerns of Reviewer 3 and revised the manuscript accordingly. Changes in the manuscript text are highlighted in red. Excerpts from the reviewers' critiques are indicated below in *italics*.

Reviewer #1 (Remarks to the Author):

Comments were addressed. I recommend to accept.

We thank the reviewer for reviewing our work and for the positive recommendation.

Reviewer #2 (Remarks to the Author):

My concerns were well-addressed in the revised manuscript as summarized in the response to reviewers cover letter.

We thank the reviewer for reviewing our work and for the positive recommendation.

Reviewer #3 (Remarks to the Author):

This is a well-designed study of LNP-G6PC mRNA therapy for GSD-Ia. It builds on the substantial contributions of Roseman et al (Mol Ther 26:814-821, 2018) who have shown that that a single administration of LNP-G6PC mRNA corrects fasting hypoglycemia and normalized metabolic abnormalities in L-G6pc^{-/-} mice, validating the therapeutic approach. The additional findings contributed by the present manuscript is to show that repeated dosing of LNP-G6PC mRNA is safe. In the revision, the authors further showed that repeated dosing of LNP-G6PC mRNA reduced tumor incidence, a new but an expected finding. Previously, Cho et al had shown that treating L-G6pc^{-/-} mice at the tumor-developing stage with rAAV-G6PC restored hepatic G6Pase- α expression, normalized glucose homeostasis and prevented de novo HCA/HCC development (J Inherit Metab Dis 42:459-469, 2019).

We thank the reviewer for the thorough and thoughtful analysis of our study.

A remaining concern about the revised manuscript is the failure to acknowledge earlier work in the field, leaving the implication that the S298C variant and codon optimization they report are novel findings. Zhang et al (J Inherit Metab Dis 42:470-479, 2019) documented in vitro and in vivo characterization of four hG6PC constructs, hG6PC-wild-type, codon optimized (co) hG6PC, hG6PC-S298C, and co-hG6PC-S298C, and showed that the co-hG6PC-S298C is 3 to 4-fold more active than the hG6PC-wild-type construct. Therefore, the statement in the revised manuscript on page 7, paragraph 2 and the statement transmitted to the reviewer, "Furthermore, combination of the protein engineering approach with the codon-optimization modification at nucleotide level makes this study a unique contribution to the field" is misleading.

As requested by the reviewer, we have now removed this statement from the text.

[Type here]

The authors' statements in the revised manuscript on page 6, lines 12-13 and on page 7, lines 3-4 also contradicts the references they do make about the study by Zhang et al. In Results on page 6, lines 12-13, the authors stated, "This finding is consistent with previous studies by Zhang et al., which also report improvements in protein expression with the S298C substitution^{47,48}."; and on page 7, lines 3-4 the authors stated, "In addition to protein engineering, we performed codon optimization of the hG6PC mRNA sequence to further enhance protein expression and enzyme activity."

We appreciate the reviewer's concern and wish to clarify that it was not our intention to claim that the S298C variant is a novel finding of this study. Indeed, we state that this finding supports/confirms previous work. To clarify this point further in the text (page 6, lines 12-13; and page 7, 2nd paragraph), we now state: "This finding is consistent with and supports previous studies by Zhang et al., which reported similar improvements in protein expression with the S298C variant."

Specific Comments

1. Page 10, lines 14-16: the statement, "it has been reported that a 2.5-hr fast in mice is equivalent to a 6 to 8-hr fast in humans, while a 5-6-hr fast is comparable to a 16 to 18-hr fast in humans⁵⁸" is misleading. Jensen et al (reference 58) only stated that "Fasting for 5-6 h instead of overnight (16-18 h) might offer a better comparison to humans, who are usually fasted overnight before similar tests." While the L-G6pc^{-/-} mice manifest most clinical manifestations of human GSD-1a, they exhibit normoglycemia in the fed state and can survive prolonged fasting because the induction of glucose production from the kidney and intestine has rescued the L-G6pc^{-/-} mice from suffering hypoglycemic seizures, the hallmark of human GSD-1a. The authors should take precaution to extrapolate fasting data obtained using L-G6pc^{-/-} mice to human GSD-1a patients.

We thank the review for the comment pertaining to the fasting interval comparison between L-G6pc^{-/-} mice and human GSD1a patients and have revised the text accordingly (Page 10, lines 12-15). The reviewer is correct to point out that, unlike human patients, the L-G6pc^{-/-} mice "exhibit normoglycemia in the fed state and can survive prolonged fasting because the induction of glucose production from the kidney and intestine has rescued the L-G6pc^{-/-} mice from suffering hypoglycemic seizures, the hallmark of human GSD-1a." Despite these obvious limitations, as indicated by the reviewer, this model manifests most clinical manifestations of human GSD1a and has thus, been successfully used by many laboratories (including us in the present study) to examine effect of various therapies on blood glucose levels during the fasted state.

2. Page 15, lines 9-11: the statement, "While encouraging, the gene therapy approach was not able to abrogate pre-existing tumors due to lack of expression of the virus in the adenoma lesions⁶⁸." Reference 68 was incorrectly cited. The reviewer believes Cao et al were referencing the study by Cho et al. (J Inherit Metab Dis 42:459-469, 2019) showing that treating L-G6pc^{-/-} mice at the tumor-developing stage with rAAV-G6PC restored hepatic G6Pase- α expression, normalized glucose homeostasis, and prevented de novo HCA/HCC development. Cho et al further showed that gene therapy could not restore G6Pase- α expression in the HCA/HCC lesions and failed to abrogate the pre-existing tumors. The conclusion was reached via examination of G6Pase- α expression in the tumor by immunohistochemical analysis. Unless Cao et al examined G6Pase- α activity in the tumor of LNP-G6PC mRNA-treated L-G6pc^{-/-} mice by immunohistochemical analysis, the authors cannot conclude, "The expectation is that GSD1a mRNA therapy may, not only prevent de novo HCA/HCC development at the tumor developing stage, but also potentially reduce any pre-existing tumor burden."

[Type here]

We agree with the reviewer and now revised the sentence on page 15, 2nd paragraph, to clarify that our statement is purely speculative based on the findings recently reported by Cho *et al.* (*J Inherit Metab Dis* 42:459-469, 2019). We now state: *“While purely speculative, the expectation is that GSD1a mRNA therapy may, not only prevent de novo HCA/HCC development at the tumor developing stage, but also potentially reduce any pre-existing tumor burden. Given the potential of GSD1a mRNA therapy to reduce pre-existing tumors, further studies in older mice with pre-existing adenomas are warranted.”* Finally, as suggested by the reviewer, we include both the Lee *et al.* and Cho *et al.* references (Page 15, 2nd paragraph, references 68 and 69).

[Type here]

Reviewers' Comments:

Reviewer #3:

Remarks to the Author:

My concerns have been addressed and I have no additional comments.

Response to the reviewers' comments

We thank the reviewers for their comments on our revised manuscript. We are pleased to learn that our manuscript is in principle accepted for publication after suitable revisions. We have also addressed the editorial comments and revised the manuscript accordingly (response to editorial comments are provided separately in the editorial request document). Changes in the main manuscript text are highlighted in **red**. Excerpts from the reviewers' critiques are indicated below in *italics*.

Reviewer #3 (Remarks to the Author):

Reviewer #3 (Remarks to the Author):

My concerns have been addressed and I have no additional comments.

We thank the reviewer for reviewing our work and are happy to learn that all concerns have been addressed.

[Type here]